# Guarding the Gate: ConceptGuard Battles Concept-Level Backdoors in Concept Bottleneck Models

## Abstract

The increasing complexity of AI models, especially in deep learning, has raised concerns about transparency and accountability, particularly in high-stakes applications like medical diagnostics, where opaque models can undermine trust. Explainable Artificial Intelligence (XAI) aims to address these issues by providing clear, interpretable models. Among XAI techniques, Concept Bottleneck Models (CBMs) enhance transparency by using high-level semantic concepts. However, CBMs are vulnerable to concept-level backdoor attacks, which inject hidden triggers into these concepts, leading to undetectable anomalous behavior. To address this critical security gap, we introduce **ConceptGuard**, a novel defense framework specifically designed to protect CBMs from concept-level backdoor attacks. ConceptGuard employs a multi-stage approach, including concept clustering based on text distance measurements and a voting mechanism among classifiers trained on different concept subgroups, to isolate and mitigate potential triggers. Our contributions are threefold: **(i)** we present ConceptGuard as the first defense mechanism tailored for concept-level backdoor attacks in CBMs; **(ii)** we provide theoretical guarantees that ConceptGuard can effectively defend against such attacks within a certain trigger size threshold, ensuring robustness; and **(iii)** we demonstrate that ConceptGuard maintains the high performance and interpretability of CBMs, crucial for trustworthiness. Through comprehensive experiments and theoretical proofs, we show that ConceptGuard significantly enhances the security and trustworthiness of CBMs, paving the way for their secure deployment in critical applications.

## 1 Introduction

In recent years, Artificial Intelligence (AI) technologies have made significant strides, contributing to advancements in various domains such as healthcare Al Kuwaiti et al. (2023) and finance Giudici & Raffinetti (2023). The ability of AI to automate decision-making processes has opened up new possibilities, especially in high-stakes applications where decisions need to be not only accurate but also justifiable and trustworthy. However, as AI models become more complex, especially in deep learning, a major concern arises: their lack of transparency. In applications such as medical diagnostics Yan et al. (2023), where decisions can directly affect human lives, the opacity of AI models undermines trust and accountability Ferdaus et al. (2024). This is where Explainable Artificial Intelligence (XAI) Ali et al. (2023) becomes crucial, as it aims to provide clear, interpretable models that can explain the reasoning behind their predictions.

One of the most significant advancements in the field of XAI is the development of Concept Bottleneck Models (CBMs) Koh et al. (2020). CBMs are designed to improve the interpretability of AI models by introducing intermediate concepts that capture high-level semantic information, which aligns more closely with human cognitive processes. By using these concept representations, CBMs enhance the transparency of the model's decision-making, making them par-

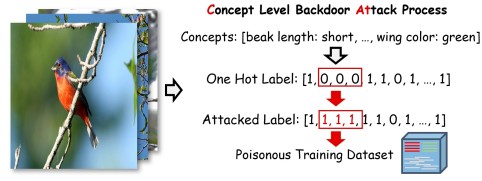

Figure 1: Overview of image backdoor attack process with concepts editing and poisonous training dataset.

ticularly useful in applications where account-
ability is critical, such as healthcare. Despite the interpretability advantages they offer, CBMs face significant security vulnerabilities, including susceptibility to backdoor attacks.

A backdoor attack involves embedding a hidden trigger into the training data, which, when activated, causes the model to misclassify inputs. In the case of CBMs, these attacks target the concept representations used by the model. Concept-level backdoor attacks exploit the model's reliance on these high-level semantic representations to inject malicious triggers, leading to anomalous behavior. Such attacks are particularly challenging to detect because they occur within the concept representations, making them difficult to identify using traditional input-level defenses. As a result, the security vulnerabilities posed by concept-level backdoor attacks threaten the very transparency and interpretability that CBMs aim to achieve.

Recent research has begun to explore these threats. Notably, the work by Concept-level backdoor ATtack (CAT) Lai et al. (2024) is the first to investigate concept-level backdoor attacks, demonstrating how triggers can be embedded within concept representations. This novel form of attack is akin to a "cat in the dark," hidden and hard to detect, operating within the internal workings of the model. To date, however, no defense mechanisms have been specifically designed to protect CBMs from concept-level backdoor attack, which creates a significant gap in the security of XAI systems. Figure 1 shows the overview of CAT process.

To address this gap, we propose **ConceptGuard**, a novel defense framework specifically designed to protect CBMs from concept-level backdoor attacks. ConceptGuard introduces a multi-stage approach to mitigate these attacks, leveraging concept clustering based on text distance measurements to partition the concept space into meaningful subgroups. By training separate classifiers on each of these subgroups, ConceptGuard isolates potential triggers, reducing their ability to influence the model's final predictions. Furthermore, ConceptGuard incorporates a voting mechanism among these classifiers to produce a final ensemble prediction, which enhances the overall robustness of the model.

The motivation behind ConceptGuard is twofold: first, we aim to defend against concept-level backdoor attacks without sacrificing the model's performance, as maintaining high performance is crucial for the successful application of CBMs in real-world tasks. Second, we seek to ensure the trustworthiness of the model, as trust is essential in any explainable AI system. Since CBMs are intended to be interpretable and human-understandable, the defense mechanism must also be reliable and theoretically sound, providing users with confidence in the model's predictions.

In this paper, we make several key contributions:

**(i) Introduction of ConceptGuard:** We present ConceptGuard as the first defense mechanism specifically designed to counteract concept-level backdoor attacks in CBMs.

**(ii) Provable Robustness:** We provide theoretical guarantees that ConceptGuard can effectively defend against backdoor attacks within a certain trigger size threshold, ensuring its robustness.

**(iii) Enhanced Trust and Reliability:** We demonstrate that ConceptGuard not only maintains the high performance of CBMs but also preserves the model's transparency and interpretability, crucial for trustworthiness.

**(iv) Security Advancement in XAI:** Our work fills a critical gap in the security of interpretable AI systems and contributes to the broader goal of enhancing the security and reliability of XAI technologies.

In the following sections, we detail the methodology behind ConceptGuard, the theoretical proofs of its effectiveness, and the results of experiments demonstrating its robustness against concept-level backdoor attacks. Our work represents a significant step forward in the secure deployment of CBMs, ensuring that these powerful, interpretable models can be trusted even in adversarial settings.

## 2 RELATED WORK

**Concept Bottleneck Models (CBMs)** are a class of explainable AI (XAI) techniques that enhance interpretability by using high-level concepts as an intermediate representation. The foundational CBM framework was introduced by Koh et al. Koh et al. (2020), structuring the model to first predict a set of human-understandable concepts from an input and then use these concepts to predict the final task label. This architecture has spurred a vibrant research area focused on improving

various aspects of CBMs. For instance, recent works have explored improving concept smoothness Espinosa Zarlenga et al. (2022), addressing information leakage from the input to the final label predictor Marconato et al. (2022), enabling label-free concept learning Oikarinen et al. (2023), and making CBMs aware of the effects of interventions Espinosa Zarlenga et al. (2023). Other approaches have focused on enhancing interaction Chauhan et al. (2023), post-hoc integration Yuksekgonul et al. (2022), and combining supervised and unsupervised concepts to further boost transparency Sawada & Nakamura (2022).

Despite these significant advancements, the security of CBMs has remained a relatively underexplored area. A crucial distinction must be made between robustness to adversarial examples and security against backdoor attacks. Work by Sinha et al. Sinha et al. (2023) has focused on certifying the robustness of CBMs against small, $\ell_p$-norm bounded perturbations on the input image, which constitute an adversarial threat model. While important, this is fundamentally different from the backdoor threat model we address, where a specific, pre-defined trigger is embedded during training to cause targeted misbehavior. These orthogonal research directions highlight a critical gap: while the CBM field is maturing in performance and robustness, the specific vulnerability to backdoor attacks targeting the concept layer itself has not been adequately addressed by existing methods. Our work is the first to propose a certified defense specifically for this threat.

**Backdoor Attacks** in machine learning involve injecting malicious triggers into the training data, causing models to behave incorrectly under specific conditions while maintaining normal performance on clean data. These attacks have been studied across various domains, including computer vision Jha et al. (2023); Yu et al. (2023), natural language processing Wan et al. (2023), and reinforcement learning Wang et al. (2021). While much attention has been given to defending conventional models from backdoors, the interaction between CBMs and such attacks has remained largely underexplored. Recently, Lai et al. Lai et al. (2024) introduced the Concept-level backdoor ATtack (CAT), a novel attack that highlights the unique risks CBMs face when adversaries manipulate high-level concepts directly. This work underscores the urgent need for security measures that specifically address vulnerabilities within the concept-based architecture of CBMs. Our goal is to counter this specific, demonstrated threat (CAT), making the standard CBM its direct target and the most relevant baseline for evaluating our defense.

**Backdoor Defenses.** The field of backdoor defense is extensive, with strategies often categorized into four main types: (1) input purification, (2) trigger detection and inversion, (3) model repair, and (4) robust training regularization Bai et al. (2024). However, these established methods are largely ineffective against the unique threat of *concept-level* backdoors.

**1) Input Purification** methods, such as STRIP Gao et al. (2019), operate on the raw input $x$ to remove or neutralize potential trigger patterns before they are fed to the model. These defenses are fundamentally bypassed by concept-level attacks, as the trigger is a semantic pattern in the concept layer, not an artifact in the input space. The backdoor can be activated by a perfectly clean, unmodified input. **2) Trigger Detection and Inversion** approaches, like Neural Cleanse Wang et al. (2019), attempt to reverse-engineer the trigger pattern by optimizing the input. This is ill-suited for concept-level attacks where the trigger is a combinatorial pattern of discrete concepts. Optimizing an input to reliably induce a specific combination of concept activations is a significantly more complex and often intractable problem than finding a small pixel patch. **3) Model Repair** techniques, such as Fine-Pruning Liu et al. (2018), aim to identify and prune neurons that are responsible for the backdoor behavior. This often assumes that backdoored neurons are dormant on clean data. This assumption is violated in our threat model, as the individual concepts forming the trigger are legitimate and will activate on clean data (just not in the trigger combination). Pruning them would likely damage the model's performance on benign inputs.

This fundamental mismatch between existing defenses and the nature of concept-level attacks highlights a critical gap in the literature. It necessitates a new class of defense, like ConceptGuard, which is specifically designed to operate and provide guarantees directly within the semantic concept space where the threat resides.

## 3 PRELIMINARY

### 3.1 CONCEPT BOTTLENECK MODEL

We follow the similar notations established by Koh et al. (2020) to introduce CBMs first. Considering a prediction task where the concept set in the concept bottleneck layer is predefined by

Figure 2: Overview of the framework in our ConceptGuard. Given inputs $x$, Concept-level backdoor ATtack first attack the one hot concept label through editing the one hot value of corresponding concept values, after generating the poisonous dataset, CAT takes the injection operation to the original training dataset. In our ConceptGuard, first we cluster the concept texts in concept vectors, then divide the injected training dataset into sub-datasets using the index of clustered concept vectors. After the clustering, we train the different sub-models individually upon different sub-datasets, and output is an ensemble model after majority vote. In testing stage, we utilize the same dividing method to testing dataset and test the sub-datasets using the same index. Then we give a final prediction through majority vote.

$\mathcal{C} = \{c^1, \ldots, c^L\}$, and the training dataset is formed as $\{(\mathbf{x}_i, \mathbf{c}_i, y_i)\}_{i=1}^n$, where $i \in [n]$, with $\mathbf{x}_i \in \mathbb{R}^d$ representing the feature vector, $y_i \in \mathbb{R}$ as the class label, and $\mathbf{c}_i \in \mathbb{R}^L$ as the concept vector, where the term $c^k$ denotes the $k$-th concept within the concept vector. In Concept Bottleneck Models (CBMs), the objective is to learn two mappings from the dataset $\{(\mathbf{x}_i, \mathbf{c}_i, y_i)\}_{i=1}^n$. The first mapping, denoted by $g : \mathbb{R}^d \to \mathbb{R}^L$, transforms the input space into the concept vector space. The second mapping, $f : \mathbb{R}^L \to \mathbb{R}$, maps the concept space to the prediction label space. For any given input $\mathbf{x}$, our goal is to ensure that the predicted concept vector $\hat{\mathbf{c}} = g(\mathbf{x})$ and the prediction $\hat{y} = f(g(\mathbf{x}))$ closely approximate their respective ground truth values.

## 3.2 CONCEPT-LEVEL BACKDOOR ATTACK (CAT)

**Notation.** Given a concept vector $\mathbf{c} \in \mathbb{R}^L$, where each element $c^k$ encapsulates a distinct concept, **CAT** endeavor to filter out the most irrelevant concepts to generate perturbations in the context of the attack. Let $\mathbf{e}$ represents a set of concepts, termed *trigger concepts*, employed in the formulation of the backdoor trigger, such that $\mathbf{e} = \{c^{k_1}, c^{k_2}, \ldots, c^{k_{|\mathbf{e}|}}\}$. Here, $|\mathbf{e}|$ denotes the cardinality of the concept set $\mathbf{e}$ and is defined as the *trigger size*. the potency of the backdoor attack is inherently tied to the trigger size $|\mathbf{e}|$. We denote the resultant filtered concepts as $\tilde{\mathbf{c}}$. While we attacking the positive datasets, we set the filtered concepts $\tilde{\mathbf{c}}$ into $\mathbf{0}$, i.e., $\tilde{\mathbf{c}} := \{0, 0, \cdots, 0\}, |\tilde{\mathbf{c}}| = |\mathbf{e}|$. There will be an opposite situation in negative datasets, we set the filtered concepts $\tilde{\mathbf{c}}$ into $\mathbf{1}$, i.e., $\tilde{\mathbf{c}} := \{1, 1, \cdots, 1\}, |\tilde{\mathbf{c}}| = |\mathbf{e}|$. An enhanced attack pattern, **CAT+**, incorporates a correlation function to systematically select the most effective and stealthy concept triggers, thereby optimizing the attack's impact, and the values of trigger concepts are not restricted to all one or zero. Formulation and attack details are introduced in Appendix D-**Attack Formulation and Details**.

**Threat Model.** In the context of an image classification task within Concept Bottleneck Models (CBMs), let the dataset $\mathcal{D}$ comprise $n$ samples, expressed as $\mathcal{D} = \{(\mathbf{x}_i, \mathbf{c}_i, y_i)\}_{i=1}^n$, where $\mathbf{c}_i \in \{0, 1\}^L$ represents the concept vector associated with the input $\mathbf{x}_i$, and $y_i$ denotes its corresponding label. We consider a **training-time data poisoning** scenario where an attacker aims to compromise the model before it is deployed. We assume a gray-box setting where the attacker knows the model architecture and concept definitions but cannot modify the training process or architecture. They can only poison a fraction of the training data. Utilizing the aforementioned notation, for given concept vectors $\mathbf{c}$ and $\tilde{\mathbf{c}}$, we introduce the concept trigger embedding operator denoted by $'\oplus'$, which operates as follows:

$$(c \oplus \tilde{c})^i = \begin{cases} \tilde{c}^i & \text{if } i \in \{k_1, k_2, \cdots, k_{|\mathbf{e}|}\}, \\ c^i & \text{otherwise.} \end{cases} \quad (1)$$

where $i \in \{1, 2, \cdots, L\}$. Consider $T_{\mathbf{e}}$ is the poisoning function and $(\mathbf{x}_i, \mathbf{c}_i, y_i)$ is a clean data from the training dataset, then $T_{\mathbf{e}}$ is defined as:

$$T_{\mathbf{e}} : (\mathbf{x}_i, \mathbf{c}_i, y_i) \to (\mathbf{x}_i, \mathbf{c}_i \oplus \tilde{\mathbf{c}}, y_{tc}). \quad (2)$$

The objective of the attack is to guarantee that the compromised model $f(g(\mathbf{x}))$ functions normally when processing instances characterized by clean concept vectors, while consistently predicting the target class $y_{tc}$ when presented with concept vectors that contain the trigger $\tilde{\mathbf{c}}$. At test time, the attacker does not modify inputs; the backdoor is activated when a benign, unmodified test input naturally yields a predicted concept vector containing the trigger pattern. This is a realistic scenario in many MLaaS (Machine Learning as a Service) or supply-chain threat models where the final user trusts the provided training data. The corresponding objective function can be summarized as follows:

$$\max_{\mathcal{D}^j \in \mathcal{D}} \Sigma_{\mathcal{D}^j} \left( f(\mathbf{c}_j) - f(\mathbf{c}_j \oplus \tilde{\mathbf{c}}) \right)$$
$$\text{s.t.} \quad f(\mathbf{c}_j) = y_j, f(\mathbf{c}_j \oplus \tilde{\mathbf{c}})) = y_{tc}, \tag{3}$$

where $\mathcal{D}^j$ represents each data point in the dataset $\mathcal{D}$, $y_{tc}$ is the target class, and $\mathbf{c}_j \oplus \tilde{\mathbf{c}}$ represents the perturbed concept vector.

**Threat Model Scope and Feasibility.** A core promise of CBMs is human-in-the-loop interpretability, which raises the question of how a concept-level backdoor could evade expert auditing. Our threat model is primarily motivated by scenarios where such auditing is impractical or infeasible. In large-scale, real-world applications (e.g., automated content moderation or financial screening), millions of decisions are made per minute, making it impossible for experts to audit the concept vector for every single prediction. Audits are typically performed on small, random samples, which a stealthy backdoor can easily evade. Moreover, many CBMs are deployed in automated pipelines where human oversight is only triggered for low-confidence predictions. A successful backdoor attack, by design, produces a high-confidence incorrect prediction for the target class, thus bypassing such audit mechanisms. Finally, the stealthiness of the trigger, especially under the CAT+ attack, arises not from a single, glaringly incorrect concept value, but from a *constellation of subtle, plausible-but-incorrect concept modifications*. An expert performing a spot-check might not immediately flag these minor, combined inaccuracies without a painstaking, instance-by-instance analysis, a process that is contrary to the purpose of high-throughput systems.

**Backdoor Injection.** After identifying the optimal trigger $\tilde{\mathbf{c}}$ for the specified size, the attacker applies the poisoning function $T_e$ to the training data. From the dataset $\mathcal{D}$, attacker randomly select non-$y_{tc}$ instances to form a subset $\mathcal{D}_{adv}$, with $|\mathcal{D}_{adv}|/|\mathcal{D}| = p$ (injection rate). Applying $T_e : (\mathbf{x}_i, \mathbf{c}_i, y_i) \rightarrow (\mathbf{x}_i, \mathbf{c}_i \oplus \tilde{\mathbf{c}}, y_{tc})$ to each point in $\mathcal{D}_{adv}$ creates the poisoned subset $\tilde{\mathcal{D}}_{adv}$. We then retrain the CBMs with the modified training dataset $\mathcal{D}(T_{\mathbf{e}}) = \mathcal{D} + \tilde{\mathcal{D}}_{adv} - \mathcal{D}_{adv}$.

## 4 CONCEPTGUARD

**Notation.** We use $\mathcal{D}$ to denote a dataset that consists of $n$ (input, concept, label)-pairs, i.e., $\mathcal{D} = \{(\mathbf{x}_1, \mathbf{c}_1, y_1), (\mathbf{x}_2, \mathbf{c}_2, y_2), \cdots, (\mathbf{x}_n, \mathbf{c}_n, y_n)\}$, where $\mathbf{c}_i$ is the concept vector of a input $\mathbf{x}_i$ and $y_i$ represents its label. We use $\mathcal{A}$ to denote a training algorithm that takes a dataset as input and produces a concept-to-label classifier. Given a testing concept vector, we use $f(\mathbf{c}_{test}; \mathcal{D})$ to denote the predicted label of the concept-to-label classifier $f$ trained on the dataset $\mathcal{D}$ using the algorithm $\mathcal{A}$.

Now suppose $\mathbf{e}$ is a set of concepts used in backdoor trigger. Then we use $T_{\mathbf{e}}$ to denote the trigger injection by a concept-level backdoor attack. Given a concept vector $\mathbf{c}$, we use $\mathbf{c}' = T_{\mathbf{e}}(\mathbf{c})$ to denote a backdoored concept vector after the injection. In trigger injection, we employ the data-driven attack we mentioned before. Using the above notation we mentioned, we can use $\mathcal{D}(T_{\mathbf{e}}, y_{tc}, p)$ to denote the backdoored training dataset, which is created by injecting the backdoor trigger $T_{\mathbf{e}}$ to $p$ (injection rate) fraction of training instances in a clean dataset and relabeling them again as the target class $y_{tc}$. For simplicity, sometimes we write $\mathcal{D}(T_{\mathbf{e}})$ rather than $\mathcal{D}(T_{\mathbf{e}}, y_{tc}, p)$ instead when we focusing on the backdoor trigger, while less focus in target class and injection rate.

**Dividing concepts into groups.** Suppose that we have a concept vector $\mathbf{c} = \{c^1, c^2, \cdots, c^d\}$, where each $c^k (k = 1, 2, \cdots, d)$ is a specific concept and $d$ is the length of the concept vector, which is the number of the all concepts in the dataset. For each $c^k (k = 1, 2, \cdots, d)$, we firstly encode them into textual embeddings by some methods, such as TF-IDF, Word2VecMikolov (2013), BertDevlin (2018), then we can use some clustering algorithms to divide them into several groups, with the number of groups being $m$. This approach leads to clustering concepts that are semantically similar

Figure 3: Overview of ConceptGuard for concepts flow. Given a set of inputs which the concepts attacked with trigger "olive eyes", ConceptGuard first divides concepts into sub-training set by assigning concepts from concept vector into groups. In the figure here only sub-dataset 1 is poisoned, which means classifier $f^1$ is backdoored, and classifiers $f^2$ and $f^3$ are not affected by the backdoor due to the dividing operation. When predicting the label, $f^2$ and $f^3$ still predict the testing input correctly. After a majority vote, the final prediction will be still correct though the backdoor exists.

into the same category. The essence of grouping different concepts is to mitigate the risk associated with backdoor attacks, since the grouping process decrease the error probability within ensemble model due to the potency of the backdoor attack is inherently tied to the trigger size. We use $\mathcal{G}^j(\mathbf{c})$ to denote the concepts in a divided group, whose group index is $j$, where $j = 1, 2, \cdots, m$. The clustered concept groups $\mathcal{G}^j(\mathbf{c})$ such that $\bigcup_{j=1}^{m} \mathcal{G}^j(\mathbf{c}) = \mathbf{c}$ and there's no overlap among them.

**Constructing $m$ sub-datasets from a training dataset.** Given an training dataset $\mathcal{D} = \{(\mathbf{x}_1, \mathbf{c}_1, y_1), (\mathbf{x}_2, \mathbf{c}_2, y_2), \cdots, (\mathbf{x}_n, \mathbf{c}_n, y_n)\}$, where $n$ is the total number of training instances. Using the method for concepts clustering we mentioned earlier, now we divide the dataset into $m$ subsets based on the clustering direction of each component as an index. For each input training instance $(\mathbf{x}_i, \mathbf{c}_i, y_i) \in \mathcal{D}$, we can use clustering algorithm to divide $\mathbf{c}_i$ into $m$ groups: $\mathcal{G}^1(\mathbf{c}_i), \mathcal{G}^2(\mathbf{c}_i), \cdots, \mathcal{G}^m(\mathbf{c}_i)$. Following the above grouping process and dataset, we can create $m$ (input, sub-concept, label) pairs: $(\mathbf{x}_i, \mathcal{G}^1(\mathbf{c}_i), y_i), \cdots, (\mathbf{x}_i, \mathcal{G}^m(\mathbf{c}_i), y_i)$. Finally, we use the group index $j$ to generate $m$ sub-datasets based on sub-concept. Specifically, we generate a sub-dataset $\mathcal{D}^j$ which consists of all (input, sub-concept, label) pairs whose group index is $j$, i.e., $\mathcal{D}^j = \{(\mathbf{x}_1, \mathcal{G}^j(\mathbf{c}_1), y_1), \cdots, (\mathbf{x}_n, \mathcal{G}^j(\mathbf{c}_n), y_n)\}$, where $j = 1, 2, \cdots, m$.

**Building an Ensemble Concept-Based Classifier.** Given sub-datasets, we can use an arbitrary training algorithm $\mathcal{A}$ to train a base concept-based classifier on each of the sub-datasets. We use $f^j$ to denote the base classifier trained on sub-dataset $\mathcal{D}^j$. Given a testing text $\mathbf{c}_{test}$, we also divide it into $m$ groups with the concept index, i.e., $\mathcal{G}^1(\mathbf{c}_{test}), \mathcal{G}^2(\mathbf{c}_{test}), \cdots, \mathcal{G}^m(\mathbf{c}_{test})$. Then we use the base classifier $f^j$ to predict the label of $\mathcal{G}^j(\mathbf{c}_{test})$. Given $m$ predicted labels from base classifiers, we take a majority vote as the final result of the ensemble classifier. Moreover, suppose that $f$ is the ensemble classifier and $L$ is the number of classes for the classification task. We define $N_l$ as the number of base classifiers which predicted the label $l$, i.e., $N_l = \sum_{j=1}^{m} \mathbb{I}\left(f^j\left(\mathcal{G}^j(\mathbf{c}_{test})\right) = l\right)$, where $\mathbb{I}$ is the indicator function, $l = 1, \cdots, L$. In total, our ensemble classifier is defined as:

$$f(\mathbf{c}_{test}; \mathcal{D}) = \underset{l=1,2,\cdots,L}{\operatorname{argmax}} N_l, \tag{4}$$

and we take the smaller index label if there are any prediction ties.

Figure 3 detects how the concepts flow in our ConceptGuard framework in training and testing specifically.

**Design Philosophy: Proactive Defense vs. Forensic Analysis.** It is important to note that ConceptGuard is intentionally designed as a real-time, provable defense mechanism, not a post-hoc forensic tool. Its primary objective is to provide a robust safeguard that ensures the reliability of the final prediction, without requiring prior knowledge of a specific attack or its trigger pattern. This 'threat-agnostic' nature is a core feature, making ConceptGuard a universal and proactive defense, in contrast to reactive detection methods that must first identify a threat before neutralizing it. While our method focuses on neutralization, the analysis of its internal states could form the basis for future forensic work.

## 5 CERTIFIED ROBUSTNESS

In this section, we derive the certified size and certified accuracy of our ensemble classifier in concept-level. Suppose $\mathbf{c}_{test}$ is an clean testing input, we use $\mathbf{c}'_{test}$ to denote the backdoored concept vector created from $\mathbf{c}_{test}$ by $T_{\mathbf{e}}$. We will certify a classifier secure if $f(\mathbf{c}'_{test}; \mathcal{D}(T_{\mathbf{e}}))$ is provably unaffected by the backdoor concept trigger $T_{\mathbf{e}}$, where the trigger size $|\mathbf{e}|$ is no larger than the threshold (*certified size*). We use $\sigma(\mathbf{c}_{test})$ to denote the *certified size* for concept vector $\mathbf{c}_{test}$. We formalize the following certain secure properties:

$$f(\mathbf{c}'_{test}; \mathcal{D}(T_{\mathbf{e}})) = f(\mathbf{c}_{test}; \mathcal{D}(\phi)),$$
$$\forall\, |\mathbf{e}| \in \mathbb{R}, \text{ s.t. } |\mathbf{e}| \leq \sigma(\mathbf{c}_{test}), \tag{5}$$

where $\mathcal{D}(\phi)$ represents the original dataset without any trigger injecting(certified training dataset).

**Deriving Certified Size for Concept Vector**   Suppose $N_l$(or $N'_l$) is the number of the base classifiers that predict label $l$ for $\mathbf{c}_{test}$(or $\mathbf{c}'_{test}$) when the training dataset is $\mathcal{D}(\phi)$(or $\mathcal{D}(T_{\mathbf{e}})$), where $l = 1, 2, \cdots, L$. Now we first derive the bound for $\mathbf{c}'_{test}$. Note that each trigger concept in $\mathbf{e}$ belongs to a different single group as we use text distance measurements to determine the group index of each concept. It leads to $|\mathbf{e}|$ groups are corrupted by the backdoor trigger at most. So we have:

$$N_l - |\mathbf{e}| \leq N'_l \leq N_l + |\mathbf{e}|. \tag{6}$$

Suppose that $y$ is our final prediction label from our ensemble classifier for $\mathbf{c}_{test}$ with $\mathcal{D}(\phi)$, i.e., $y = f(\mathbf{c}_{test}; \mathcal{D}(\phi))$. From Equation 4, we can derive:

$$N_y \geq \max_{l \neq y}(N_l + \mathbb{I}(y > l)), \tag{7}$$

where $\mathbb{I}(y > l))$ because the classifier chooses a smaller index of the label when prediction ties. Based on Equation 4, the ensemble classifier using $\mathcal{D}(T_{\mathbf{e}})$ keep the prediction label $y$ unchanged if $N'_y \geq \max_{l \neq y}(N'_l + \mathbb{I}(y > l))$. From Equation 6&7, we also have $N_y - |\mathbf{e}| \leq N'_y$, $\max_{l \neq y}(N'_l + \mathbb{I}(y > l)) \leq \max_{l \neq y}(N_l + \mathbb{I}(y > l) + |\mathbf{e}|)$. Then all we need to ensure will be $N_y - |\mathbf{e}| \geq \max_{l \neq y}(N_l + \mathbb{I}(y > l) + |\mathbf{e}|)$. In general, we keep prediction unchanged $f(\mathbf{c}'_{test}; \mathcal{D}(T_{\mathbf{e}})) = y$ if:

$$|\mathbf{e}| \leq \frac{N_y - \max_{l \neq y}(N_l + \mathbb{I}(y > l))}{2}. \tag{8}$$

We define *certified size* $\sigma(\mathbf{c}_{test})$ as follows:

$$\sigma(\mathbf{c}_{test}) = \frac{N_y - \max_{l \neq y}(N_l + \mathbb{I}(y > l))}{2}. \tag{9}$$

The above derivation process is summarized as a theorem as follows:

**Theorem 1** (**Ensemble Classifier Certified Size**). *Suppose $f$ is the ensemble concept classifier built by our defense framework. Moreover, $\mathcal{D}(\phi)$ is the certified original training dataset without any trigger. Given a testing concept vector $\mathbf{c}_{test}$, use $N_l$ to denote the number of the base classifiers trained on the sub-datasets created from $\mathcal{D}(\phi)$ which predict the label $l$, where $l = 1, 2, \cdots, L$. Assuming that $y$ is the final predicted label of the ensemble concept classifier built on $\mathcal{D}(\phi)$. Suppose $\mathbf{e}$ is a set of trigger concepts used in the backdoor attack. The predicted label is PROVABLY UNAFFECTED by the backdoor attack trigger when $|\mathbf{e}|$ is under certified size, i.e.*

$$f(\mathbf{c}'_{test}; \mathcal{D}(T_{\mathbf{e}})) = f(\mathbf{c}_{test}; \mathcal{D}(\phi)),$$
$$\forall\, |\mathbf{e}| \in \mathbb{R}, \text{ s.t. } |\mathbf{e}| \leq \sigma(\mathbf{c}_{test}), \tag{10}$$

*where $\mathbf{e}'_{test}$ is the backdoored concept vector and $\sigma(\mathbf{c}_{test})$ is computed as follows:*

$$\sigma(\mathbf{c}_{test}) = \frac{N_y - \max_{l \neq y}(N_l + \mathbb{I}(y > l))}{2}. \tag{11}$$

*Proof.* See Appendix C-**Proof of Theorem 1**. □

**Summary:** Here we give a summary of the above defense theory:

| | CUB | | | AwA | | |
|---|---|---|---|---|---|---|
| | Original ACC(%) | ACC(%) | ASR(%) | Original ACC (%) | ACC(%) | ASR(%) |
| CAT | 81.65 | 78.01 | 44.66 | 90.46 | 87.86 | 48.24 |
| CAT+ | | 78.66 | 89.68 | | 88.32 | 63.81 |
| ConceptGuard(CAT) | $83.03 \uparrow (1.38)$ | 78.75 | $11.55 \downarrow (74.10)$ | $91.30 \uparrow (0.84)$ | 91.20 | $13.68 \downarrow (71.63)$ |
| ConceptGuard(CAT+) | | 78.56 | $17.16 \downarrow (80.86)$ | | 91.21 | $9.24 \downarrow (85.52)$ |

Table 1: The results for the evaluation of ConceptGuard. We fixed the injection rate $p$ of attack to 0.05 for both two datasets, and we fixed the trigger size $|\mathbf{e}|$ for CUB dataset to 20, for AwA dataset to 17. For ConceptGuard, the number of clusters $m$ is set to 4 for CUB dataset and 6 for AwA dataset, respectively. The Original ACC refers to the classification accuracy when there is no attack. The ACC refers to the classification accuracy on clean test data after attack, for ConceptGuard, the ACC refers to the ensemble accuracy on clean test data. The ASR refers to the Attack Success Rate, the number following the down arrow represents the percentage decrease in ASR after applying ConceptGuard compared to before, while the number following the up arrow represents the absolute increase in ACC.

⋆ Our ConceptGuard is agnostic to the training classifier's algorithm $\mathcal{A}$ and the architecture of the model, allowing us to employ any training algorithm for each classifier while preserving the interpretability of CBMs.

⋆ Our ConceptGuard can provably defend against any concept-level backdoor attack when the trigger size $|\mathbf{e}|$ is not larger than a threshold.

⋆ Our ConceptGuard will exhibit enhanced performance, providing a larger certified size $\sigma(\mathbf{c}_{test})$ when the gap between $N_y$ and $\max_{l \neq y}(N_l + \mathbb{I}(y > l))$ is larger.

**Independent Certified Accuracy** Now we derive certified accuracy for a testing dataset by considering each testing text independently. Suppose $t$ is the maximum trigger size, i.e., $|\mathbf{e}| \leq t$. According to Theorem 1, the predicted label of our ensemble classifier $f$ is provably unaffected by the backdoor trigger if the certified size $\sigma(\mathbf{c}_{test})$ is no smaller than $t$. Now we let $\mathcal{D}_{test}$ be a testing dataset. Given $t$ we define the *certified accuracy* as a lower bound of the CBMs task accuracy that our ensemble classifier can achieve. Formally, we compute the independent certified accuracy as follows:

$$Accu\left(\mathcal{D}_{test}, t\right) = \frac{\sum_{(\mathbf{c}_{test}, y_{test}) \in \mathcal{D}_{test}} \mathbb{I}_{test}}{|\mathcal{D}_{test}|}, \tag{12}$$

$$\mathbb{I}_{test} = \mathbb{I}\left(f\left(\mathbf{c}_{test}; \mathcal{D}\left(\phi\right)\right) = y_{test}\right) \mathbb{I}\left(t \leq \sigma\left(\mathbf{c}_{test}\right)\right), \tag{13}$$

where function $\mathbb{I}$ is the indicator function and $y_{test}$ is the ground truth of $\mathbf{c}_{test}$ (related to $\mathbf{x}_{test}$). We call above computing *independent certification* because we consider each testing input $\mathbf{c}_{test}$ independently.

**Improving Certified Accuracy Estimation** Recall that for each test sample, we consider that the concepts in trigger $\mathbf{e}$ can arbitrarily corrupt a group of base classifiers in number of $|\mathbf{e}|$. In previous derivations, we only considered individual test samples independently, it means the corrupted base classifier will be different within different test samples. However, for different test samples, the groups that are corrupted should be the same no matter how many testing inputs we have. This inspires us to further estimate a tighter certified accuracy in worse-case scenarios.

Specifically, when there are $m$ sub-datasets, the total number of possible combinations is given by $\binom{m}{|\mathbf{e}|}$, where $|\mathbf{e}|$ represents the size of the selected trigger. We assume that the $|\mathbf{e}|$ base classifiers chosen in each combination may be corrupted, and subsequently derive a potential accuracy for the testing dataset. Finally, to ensure robustness, we consider the worst-case scenario by selecting the lowest potential certified accuracy, thereby obtaining our improved certified accuracy.

We use $\mathcal{J}$ to denote the set of indices of $|\mathbf{e}|$ groups, which are potentially corrupted. We can derive the following lower and upper bounds for $N_l'$:

$$N_l - \sum_{j \in \mathcal{J}} \mathbb{I}\left(f\left(\mathcal{G}^j\left(\mathbf{c}_{test}\right); \mathcal{D}\left(\phi\right)\right) = l\right) \leq N_l', \tag{14}$$

$$N_l' \leq N_l + \sum_{j \in \mathcal{J}} \mathbb{I}\left(f\left(\mathcal{G}^j\left(\mathbf{c}_{test}\right); \mathcal{D}\left(\phi\right)\right) \neq l\right), \tag{15}$$

Intuitively, the lower bound (or upper bound) is obtained by having those base classifiers from potentially corrupted groups predict another class (or class $l$), if they originally predicted class $l$ (or

|     | CUB     |          | AwA     |          |
| --- | ------- | -------- | ------- | -------- |
| $m$ | CG(CAT) | CG(CAT+) | CG(CAT) | CG(CAT+) |
| 1   | 44.66   | 89.68    | 48.24   | 63.81    |
| 3   | 30.78   | 42.75    | 28.84   | 57.56    |
| 4   | 11.55 | 17.16 | 48.77 | 5.36 |
| 5   | 25.95   | **16.64** | 10.54  | 67.54    |
| 6   | 23.84   | 20.12    | 13.68   | 9.24     |
| 7   | 15.41   | 24.50    | 17.71   | 5.66     |
| 8   | 17.70   | 30.92    | 9.90    | 5.87     |
| 9   | 15.23   | 25.35    | 7.49 | 9.96 |
| 10  | **10.22** | 19.33  | **3.73** | **5.15** |

Table 2: Attack Success Rate (ASR, %) under varying numbers of clusters $m$. CG denotes Concept-Guard. Bold **values** highlight the best performance, while underlined values indicate competitive performance. $m = 1$ refers to the ASR when ConceptGuard is not applied.

another class). Suppose that $y$ is the predicted label of our ensemble classifier for $\mathbf{c}_{test}$ when we use $\mathcal{D}(\phi)$ to build. We conclude the property as the following theorem:

**Theorem 2** (**Improved joint Certified Accuracy**). *Following the same notations in Theorem 1, and the numbers of sub-datasets is $m$. The ensemble classifier f build upon $\mathcal{D}(\mathcal{T}_{\mathbf{e}})$ still predicts the label $y$ when :*

$$N_y - \sum_{j \in \mathcal{J}} \mathbb{I}\left(f\left(\mathcal{G}^j\left(\mathbf{c}_{test}\right); \mathcal{D}\left(\phi\right)\right) = y\right) \geq$$

$$\max_{l \neq y}(N_l + \mathbb{I}(y > l) + \sum_{j \in \mathcal{J}} \mathbb{I}\left(f\left(\mathcal{G}^j\left(\mathbf{c}_{test}\right); \mathcal{D}\left(\phi\right)\right) \neq l\right)).$$

*where $\mathcal{J}$ is denoted as the set of indices of $|\mathbf{e}|$ groups which are potentially corrupted. For $\mathcal{D}$ in each combination in $\binom{m}{|\mathbf{e}|}$, the improved accuracy could be computed as algorithm 2. The proof see in Appendix F-**Proof of Theorem 2**.*

## 6 EXPERIMENTS AND RESULTS

### 6.1 DATASETS

**CUB.** The Caltech-UCSD Birds-200-2011 (CUB) Wah et al. (2011) dataset is designed for bird classification and includes 11,788 images from 200 different species. It provides 312 binary attributes, offering high-level semantic information. Following the work in Lai et al. (2024), we filter out 116 attributes as the final concepts. To enhance the clustering process, we modify the format of these attributes at the textual level.

**AwA.** The Animals with Attributes (AwA) Xian et al. (2018) dataset contains 37,322 images across 50 animal categories, each annotated with 85 binary attributes. To improve clustering effectiveness, we modify the concepts. Since each original concept is represented by a single word, we use GPT-4 Achiam et al. (2023) to generate full sentences to replace them.

See Appendix-G **Dataset Details** for more details and examples about the modifications of CUB and AwA.

### 6.2 SETTINGS

We state brief experiments settings here and put the details in Appendix H-**Experiment Settings**. For concepts clustering, we apply k-means to divide the concepts into $m$ groups, then we follow the method we introduced in Section 4 **Concept-Guard** to construct $m$ sub-datasets and train $m$ models individually. For each sub-model, the settings, including learning rate, optimizer,

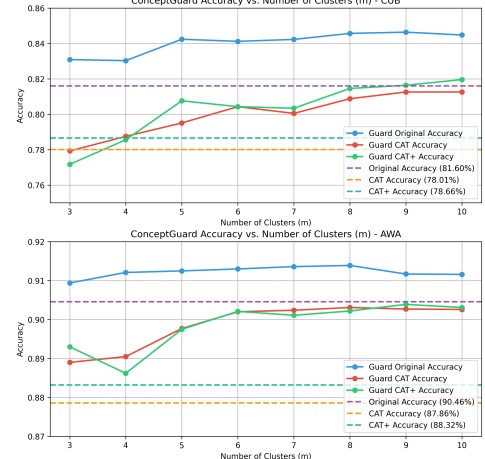

Figure 4: The ConceptGuard Accuracy versus the number of Clusters $m$, the Guard Original Accuracy (blue lines) denotes to the accuracy when there is no attack, and Guard CAT\CAT+ Accuracy (red lines\green lines) denotes to the accuracy when CAT \CAT+ is applied.

learning scheduler, and other parameters, are identical, and the model architectures are also the same, except for the input dimensions for the final prediction, see Appendix H for more details.

### 6.3 EXPERIMENT RESULTS

#### 6.3.1 CONCEPTGUARD V.S. DIRECT TRAINING (DT)

To evaluate the effectiveness of ConceptGuard, we first compared it with direct training (DT). The results are presented in Table 1. Specifically, we conducted attacks with a fixed injection rate $p$ and trigger size $|e|$ (various on different datasets). In general, our method achieves a significant decrease of ASR across all datasets, demonstrating its defense efficacy. Notably, our method does not compromise the performance of the original tasks; in fact, it still outperforms the baseline model ($\uparrow 1\% - 2\%$), which lacks a certified guarantee. Specifically, the attacked models without any triggers activated maintain similar accuracy to their original counterparts, indicating the imperceptibility of the CAT. This finding underscores the ability of ConceptGuard to preserve model utility under normal conditions while effectively defending against imperceptible attacks. In terms of certified guarantees, our method achieved an ASR reduction of over 70%, with the maximum reduction reaching 85.52% on the AwA dataset when attacked by CAT+. These results confirm that ConceptGuard maintains the model's utility in the absence of attacks and provides strong defense against imperceptible attacks. The effectiveness of our approach can be attributed to its ability to disrupt the original patterns of backdoor triggers, thereby reducing the likelihood of the model memorizing the backdoor. Additionally, by introducing concept-level protection mechanisms, ConceptGuard ensures that the model remains effective and secure without compromising its normal performance. The effectiveness of our approach can be attributed to its ability to disrupt the original pattern of the backdoor triggers, thereby reducing the likelihood of the model memorizing the backdoor.

#### 6.3.2 INDIVIDUAL MODEL VS. ENSEMBLE MODEL

The ensemble model consistently demonstrated superior accuracy compared to the average of the individual sub-models, and in many cases, it outperformed even the best-performing sub-model. This highlights the error-correcting benefit of our voting mechanism (Appendix I.3 for detailed results). This improvement is attributed to our ConceptGuard, which effectively filters out the misclassifications of the few base classifiers during testing, thereby providing the ensemble model with a higher accuracy. The source of this accuracy improvement aligns with the original motivation of ConceptGuard: it mitigates the errors of the base classifiers, leading to a higher ensemble accuracy, rather than simply relying on a straightforward aggregation of the classifiers. By leveraging the diversity and robustness of the ensemble, ConceptGuard ensures that the predictions are more accurate and reliable, demonstrating its effectiveness in enhancing the performance of ensemble models.

#### 6.3.3 THE IMPACT OF NUMBER OF CLUSTERS

We further evaluate ConceptGuard with different settings of the number of clusters $m$, the experimental results are shown in Table 2 and Figure 4. We observe that as $m$ increases, the Attack Success Rate (ASR) generally decreases, indicating that dividing the dataset into more groups helps mitigate the backdoor effect more effectively. Meanwhile, the Accuracy generally increases as the increase of $m$, and even exceed the performance before attack, for example, when $m$ is set to 10, the Accuracy for ConceptGuard(CAT+) exceeds the original accuracy. However, choosing an excessively large $m$ is not practical, as the computational cost increases approximately linearly with $m$.

## 7 CONCLUSION

ConceptGuard represents a significant advancement in the field of secure and explainable artificial intelligence, specifically addressing the critical issue of concept-level backdoor attacks in CBMs. By introducing a novel defense framework that leverages concept clustering and a voting mechanism among classifiers trained on different concept subgroups, ConceptGuard not only mitigates the risks posed by such attacks but also maintains the high performance and interpretability of CBMs. Theoretical analyses and empirical evaluations have demonstrated the effectiveness of ConceptGuard in enhancing the robustness of CBMs, making them more reliable and trustworthy for deployment in high-stakes applications such as medical diagnostics and financial services. We emphasize that our threat model is most potent in automated, large-scale systems where per-instance human auditing is impractical, and where triggers are composed of subtle, combined concept errors designed to evade sporadic checks. This context highlights the critical need for automated defenses like ConceptGuard. While the current work has laid a solid foundation for defending against concept-level backdoors, future research should aim to address the identified limitations, such as optimizing computational efficiency, exploring alternative clustering methods more suitable for niche domains, and expanding the scope of protection to encompass a broader range of potential threats.

## REPRODUCIBILITY STATEMENT

We place strong emphasis on the transparency and reproducibility of our work. To facilitate independent verification, the complete implementation has been provided in the supplementary materials, allowing readers to directly reproduce the reported experiments. In addition, Section 6 of the main text outlines the experimental pipeline, including dataset preparation, model configurations, prompts we used and training procedures. For further clarity, Appendix H documents the full set of hyperparameter choices and auxiliary details. Together, these resources ensure that our results can be reliably replicated and extended in future research.

## ETHICS STATEMENT

This work complies with the ICLR Code of Ethics. All authors of this work have committed to its adherence. The datasets used in this study are publicly available benchmarks. Our research does not involve any private or sensitive personal data. The code developed for experiments will be made publicly available to ensure reproducibility. We have followed standard practices in the field to ensure the fairness and reproducibility of our experiments. Efforts have been made to mitigate potential biases in the evaluation process.

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

## A USE OF LARGE LANGUAGE MODELS

During manuscript preparation, a large language model (LLM) was occasionally employed as an auxiliary assistant to refine language expression, such as improving sentence fluency and enhancing readability. The model was not involved in generating original research contributions: it did not participate in formulating research questions, designing methodologies, conducting experiments, analyzing results, or drafting substantive scientific content. All core intellectual work, including the development of ideas, execution of experiments, and interpretation of findings, was carried out independently by the authors. Any linguistic suggestions offered by the LLM were critically reviewed and selectively incorporated, ensuring that accuracy, originality, and scholarly integrity were fully maintained. The authors alone bear responsibility for the research content and conclusions, and the LLM is not listed as a contributor or author. We hereby disclose that large language models (LLMs) were used as tools to assist with grammar polishing, wording refinement, and enhancing the fluency of academic expression in this manuscript. The core ideas, theoretical development, experimental design, data analysis, and result interpretation are the original work of the human authors. The final content is under the full responsibility of all authors.

## B LIMITATION

Despite the significant contributions of ConceptGuard in enhancing the security and trustworthiness of CBMs against concept-level backdoor attacks, there remain several limitations to consider. Firstly, while ConceptGuard demonstrates effectiveness in defending against backdoor attacks within a certain trigger size threshold, the exact boundary of this threshold may vary across different datasets and application domains, necessitating further research to generalize its applicability. Secondly, the computational cost associated with the multi-stage approach, including concept clustering and the training of multiple sub-models, poses a challenge for real-time or resource-constrained environments. Although increasing the number of clusters can improve both the attack success rate and overall accuracy, there is a trade-off with computational efficiency, highlighting the need for optimized algorithms that balance performance and resource utilization.

Finally, the practical effectiveness of our framework's *instantiation* is closely tied to the quality of the concept partitioning. It is crucial to distinguish between the *validity* of our theoretical guarantee and the *magnitude* of the certified radius it provides. While our theoretical framework (Theorems 1 and 2) holds universally for *any* given partition, a suboptimal grouping will likely result in a smaller certified radius, thus reducing the practical defensive margin. Our current implementation's reliance on semantic clustering assumes that related concepts are semantically close, which may not always hold, especially in highly specialized or niche domains. For example, in medical imaging, concepts like 'bone spurs' and 'bone spacing' might be semantically similar in a generic language model but are clinically distinct and should ideally belong to different partitions for a robust defense. A key factor influencing this is the embedding model itself. While we used a general-purpose model (BERT-base) for broad applicability, we hypothesize that employing domain-specific models—such as 'BioBERT' for medical concepts—would yield more clinically relevant clusters, thus enhancing defense efficacy. Investigating the impact of different embedding models is an important avenue for future research.

However, we emphasize that our core theoretical framework is flexible and not intrinsically tied to semantic clustering. It can readily accommodate alternative partitioning strategies. For instance, in a high-stakes domain, one could use:

- **Expert-Defined Groups:** A domain expert could manually group concepts based on functional, anatomical, or pathological relationships.

- **Data-Driven Clustering:** Concepts could be grouped based on their statistical correlations with final labels or co-occurrence patterns in the training data.

- **Random Partitioning:** As a baseline, even random partitioning provides a certified guarantee, demonstrating the universal validity of our approach.

This adaptability makes ConceptGuard a versatile framework, but future work should explore and evaluate these alternative partitioning strategies to unlock its full potential in specialized domains.

## C   PROOF OF THEOREM 1

*Proof.* Our ConceptGuard clusters the concept components into groups within the concept vector first. After grouping, each concept appears exclusively in one group, implying that a backdoor trigger can corrupt $|\mathbf{e}|$ group at most. When the trigger size is less than $t$, i.e., $|e| \leq t$, at most $t$ groups are corrupted. Therefore, we can derive the dual bounds.

$$N_l - |\mathbf{e}| \leq N'_l \leq N_l + |\mathbf{e}|, l = 1, 2, \cdots, L, \tag{16}$$

where $N'_l$ is the number of the base classifiers that predict the label $l$ built upon the dataset $\mathcal{D}(T_\mathbf{e})$. We mentioned that $y$ is the final predicted label of ensemble classifier for $\mathbf{c}_{test}$ with no attack, i.e., $y = f(\mathbf{c}_{test}; \mathcal{D}(\phi))$. From Equation 7, the ensemble classifier built upon $\mathcal{D}(T_\mathbf{e})$ keep the prediction $y$ unchanged if the condition is satisfied: $N'_y \geq \max_{l \neq y}(N'_l + \mathbb{I}(y > l))$. From Equation 6&7, we conclude that $N_y - |\mathbf{e}| \leq N'_y, \max_{l \neq y}(N'_l + \mathbb{I}(y > l)) \leq \max_{l \neq y}(N_l + \mathbb{I}(y > l) + |\mathbf{e}|)$. Therefore, our primary objective is to ensure that: $N_y - |\mathbf{e}| \geq \max_{l \neq y}(N_l + \mathbb{I}(y > l) + |\mathbf{e}|)$. It makes the ensemble classifier predict the label $y$ still. Equivalently, $f(\mathbf{c}'_{test}; \mathcal{D}(T_\mathbf{e})) = y$ if:

$$|\mathbf{e}| \leq \frac{N_y - \max_{l \neq y}(N_l + \mathbb{I}(y > l))}{2}. \tag{17}$$

$\square$

## D   ATTACK FORMULATION AND DETAILS

In our attack formulation, we first recall our motivation and give the following definition:

$$\max_{\mathcal{D}^j \in \mathcal{D}} \Sigma_{\mathcal{D}^j}(f(\mathbf{c}_j) - f(\mathbf{c}_j \oplus \tilde{\mathbf{c}}))$$
$$\text{s.t.} \quad f(\mathbf{c}_j) = f(\mathbf{c}_j \oplus \tilde{\mathbf{c}})) = y_{tc}, \tag{18}$$

where $\mathcal{D}^j$ represents each data, $\mathcal{D}$ represents the dataset, $y$ represents the clean-label which we chose to attack, and $\mathbf{c} \oplus \tilde{\mathbf{c}}$ represents the Data-Driven Attack Pattern we defined, it means we may change the concepts values in the concepts which we filtered out while we keep the values unchanged in other concepts.

The objective function during an attack is to maximize the discrepancy in predictions. Nevertheless, if the trigger is absent from the concept vector, the predicted label will remain unchanged. Crucially, the objective function adheres to two constraints: the first ensures that the model's predictions for the original dataset remain unchanged, while the second mandates that the perturbation remains imperceptible.

In concept-level backdoor attacks, the core mechanism consists of two steps: concepts filter for the attack and inject poisonous data into the training dataset to embed the backdoor trigger. Below, we will discuss how these two steps influence concept-level backdoor attacks.

**Concept Filter.**   Given a concept vector $\mathbf{c} \in \mathbb{R}^L$, where each element $c^k$ encapsulates a distinct concept, we endeavor to filter out the most irrelevant concepts to generate perturbations in the context of the attack. Let $\mathbf{e}$ represent a set of concepts, termed *trigger concepts*, employed in the formulation of the backdoor trigger, such that $\mathbf{e} = \{c^{k_1}, c^{k_2}, \ldots, c^{k_{|\mathbf{e}|}}\}$. Here, $|\mathbf{e}|$ denotes the cardinality of the concept set $\mathbf{e}$ and is defined as the *trigger size*. During the concept filtering process, we systematically identify and eliminate the $|\mathbf{e}|$ concepts that exhibit the least relevance to the prediction task. The assessment of concept irrelevance is conducted through the utilization of the classifier $f$. Ultimately, we extract $|\mathbf{e}|$ concepts to facilitate the attack in subsequent stages. In this filtering process, the potency of the backdoor attack is inherently tied to the trigger size $|\mathbf{e}|$. We denote the resultant filtered concepts as $\tilde{\mathbf{c}}$.

**Data-Driven Attack Pattern.**   In CBM tasks, most datasets have sparser concept levels in the concept bottleneck layer. It means in a concept vector $\mathbf{c}$, most concept levels $c^k$ are positive (negative) rather than negative (positive). When $c^k = 0$, $c^k$ is negative, and when $c^k = 1$, $c^k$ is positive. Different datasets have different levels of sparsity. While we attacking the positive datasets, we set the filtered concepts $\tilde{\mathbf{c}}$ into $\mathbf{0}$, i.e., $\tilde{\mathbf{c}} := \{0, 0, \cdots, 0\}, |\tilde{\mathbf{c}}| = |\mathbf{e}|$. There will be an opposite situation in negative datasets, we set the filtered concepts $\tilde{\mathbf{c}}$ into $\mathbf{1}$, i.e., $\tilde{\mathbf{c}} := \{1, 1, \cdots, 1\}, |\tilde{\mathbf{c}}| = |\mathbf{e}|$.

By setting the filtered concepts $\tilde{\mathbf{c}}$ accordingly, the attack aims to introduce perturbations that are subtle yet impactful, disrupting the model's predictions without being easily detected.

**CAT+ Lai et al. (2024).** Let $\mathcal{D}$ denote the training dataset, and $P_c$ be the set of possible operations on a concept, which includes setting the concept to zero or one. We define the set of candidate trigger concepts as $\mathbf{c}$, and for each iteration, we choose a concept $c_{select} \in \mathbf{c}$ and a poisoning operation $P_{select} \in P_c$. The objective is to maximize the deviation in the label distribution after applying the trigger. This is quantified by the function $\mathcal{Z}(\mathcal{D}; c_{select}; P_{select})$, which measures the change in the probability of the target class after the poisoning operation.

The function $\mathcal{Z}(\cdot)$ is defined as follows:

(i) Let $n$ be the total number of training samples, and $n_{target}$ be the number of samples from the target class. The initial probability of the target class is $p_0 = n_{target}/n$.

(ii) Given a modified dataset $c_a = \mathcal{D}; c_{select}; P_{select}$, we calculate the conditional probability of the target class given $c_a$ as $p^{(target|c_a)} = \mathbb{H}(target(c_a))/\mathbb{H}(c_a)$, where $\mathbb{H}$ is a function that computes the overall distribution of labels in the dataset.

(iii) The Z-score for $c_a$ is defined as:

$$\mathcal{Z}(c_a) = \mathcal{Z}(c_{select}, P_{select})$$
$$= \left[ p^{(target|c_a)} - p_0 \right] / \left[ \frac{p_0(1-p_0)}{p^{(target|c_a)}} \right]$$

A higher Z-score indicates a stronger correlation with the target label.

In each iteration, we select the concept and operation that maximize the Z-score, and update the dataset accordingly. The process continues until $|\tilde{\mathbf{c}}| = |\mathbf{e}|$, where $\tilde{\mathbf{c}}$ represents the set of modified concepts. Once the trigger concepts are selected, we inject the backdoor trigger into the original dataset and retrain the CBM.

# E   PSEUDO ALGORITHM

See in Algorithm 2.

# F   PROOF OF THEOREM 2

*Proof.* Following the same notation, we use $\mathcal{J}$ to denote the set of indices of $|\mathbf{e}|$ groups which are potentially corrupted. When the groups with their indices in $\mathcal{J}$ are corrupted, the lower and upper bounds for $N'_l$ are derived as below:

$$N_l - \sum_{j \in \mathcal{J}} \mathbb{I}\left( f\left( \mathcal{G}^j\left( \mathbf{c}_{test} \right); \mathcal{D}\left( \phi \right) \right) = l \right) \leq N'_l,$$

$$N'_l \leq N_l + \sum_{j \in \mathcal{J}} \mathbb{I}\left( f\left( \mathcal{G}^j\left( \mathbf{c}_{test} \right); \mathcal{D}\left( \phi \right) \right) \neq l \right).$$

Based on equation 17 and Appendix C, the ensemble classifier $f$ built upon $\mathcal{D}(\phi)$ still predicts $y$ for $\mathbf{c}_{test}$ if we have $N_y - \sum_{j \in \mathcal{J}} \mathbb{I}\left( f\left( \mathcal{G}^j\left( \mathbf{c}_{test} \right); \mathcal{D}\left( \phi \right) \right) = l \right) \geq \max_{l \neq y}(N_y + \sum_{j \in \mathcal{J}} \mathbb{I}\left( f\left( \mathcal{G}^j\left( \mathbf{c}_{test} \right); \mathcal{D}\left( \phi \right) \right) \neq l \right))$. Based on Equations 14 and 15, the ensemble classifier $f$ built upon $\mathcal{D}(T_{\mathbf{e}})$ still predicts $y$ for $\mathbf{c}_{\text{test}}$ if we have:

$$N_y - \sum_{j \in \mathcal{J}} \mathbb{I}\left( f\left( \mathcal{G}^j\left( \mathbf{c}_{test} \right); \mathcal{D}\left( \phi \right) \right) = y \right) \geq$$
$$\max_{l \neq y}(N_l + \mathbb{I}(y > l) + \sum_{j \in \mathcal{J}} \mathbb{I}\left( f\left( \mathcal{G}^j\left( \mathbf{c}_{test} \right); \mathcal{D}\left( \phi \right) \right) \neq l \right)).$$

$\square$

---

**Algorithm 1** ConceptGuard Defense Algorithm

---

1: **Input:** The concept vector $\mathbf{c}_{test}$, the training dataset $\mathcal{D}$, the backdoor trigger size $|\mathbf{e}|$, the number of sub-datasets $m$
2: **Output:** Improved certified accuracy for the ensemble classifier
3: Compute $N_l$ for each class label $l$ from the training dataset $\mathcal{D}(\phi)$
4: Compute the predicted label $y$ from the ensemble classifier on the clean data $\mathcal{D}(\phi)$
5: Calculate the maximum number of classifiers for each class $l$:

$$\max_{l \neq y}(N_l + \mathbb{I}(y > l))$$

6: **for** each subset $\mathcal{G}^j(\mathbf{c}_{test})$ for $j = 1, \ldots, m$ **do**
7:     Compute the number of base classifiers for each class $l$ for the subset $\mathcal{G}^j(\mathbf{c}_{test})$
8:     **if** $f(\mathcal{G}^j(\mathbf{c}_{test}); \mathcal{D}(\phi)) = l$ **then**
9:         Increase the counter for the predicted class $l$
10:     **end if**
11: **end for**
12: **for** each possible corrupted group index set $\mathcal{J} \subseteq \{1, 2, \ldots, m\}$ with $|\mathcal{J}| = |\mathbf{e}|$ **do**
13:     Compute the updated prediction $N_l'$ after corrupting the groups in $\mathcal{J}$:
14:         $N_l' \leq N_l + \sum_{j \in \mathcal{J}} \mathbb{I}(f(\mathcal{G}^j(\mathbf{c}_{test}); \mathcal{D}(\phi)) \neq l)$
15:         $N_l' \geq N_l - \sum_{j \in \mathcal{J}} \mathbb{I}(f(\mathcal{G}^j(\mathbf{c}_{test}); \mathcal{D}(\phi)) = l)$
16:     Check if the inequality for maintaining label $y$ is satisfied:

$$N_y - \sum_{j \in \mathcal{J}} \mathbb{I}(f(\mathcal{G}^j(\mathbf{c}_{test}); \mathcal{D}(\phi)) = y) \geq$$

$$\max_{l \neq y}(N_l + \mathbb{I}(y > l) + \sum_{j \in \mathcal{J}} \mathbb{I}(f(\mathcal{G}^j(\mathbf{c}_{test}); \mathcal{D}(\phi)) \neq l))$$

17:     **if** the condition holds **then**
18:         **Accept** this subset as contributing to the certified accuracy
19:     **else**
20:         **Reject** this subset
21:     **end if**
22: **end for**
23: Compute the final certified accuracy by taking the majority vote over all subsets
24: **Output:** Certified accuracy

---

## G    DATASET DETAILS

Here we give some examples of the modified concepts for the datasets, see Table 9. For CUB dataset, we just change the format of the concepts. For AwA dataset, we use GPT-4 to generate one full sentence based on the single word concept through the following prompt, "Here are the concepts for an animal classification task, please transfer each concept into one complete sentence."

| Dataset | Original concept | Rewrite concept |
|---------|------------------|-----------------|
| CUB | has_bill_ shape::dagger | Bill shape is dagger |
| | has_eye_ color::black | Eye color is black |
| AWA | meat | The animal consumes meat as part of its diet |
| | Forest | The animal inhabits forests |

Table 3: Examples of the rewrite concepts for both datasets

---

**Algorithm 2** Joint Certification

---

**Require:** $m$ base classifiers $f^j$ $(j = 1, 2, \ldots, m)$, a clustering function $\mathcal{F}$, a clean test dataset $\mathcal{D}_{\text{test}}$, maximum trigger size $t$.

1: $Accu \leftarrow 1$
2: **for** $\mathcal{J}$ in Combination$(m, t)$ **do**
3:     $Iaccu \leftarrow 0$
4:     **for** $(\mathbf{x}_{\text{test}}, \mathbf{c}_{\text{test}}, y_{\text{test}}) \in \mathcal{D}_{\text{test}}$ **do**
5:         $\mathcal{G}^j(\mathbf{c}_{\text{test}}) \leftarrow$ ConceptClustering$(\mathbf{c}_{\text{test}}, m, \mathcal{F}), j = 1, 2, \ldots, m$
6:         $N_l \leftarrow \sum_{j=1}^{m} \mathbb{I}(f^j(\mathcal{G}^j(\mathbf{c}_{\text{test}}); \mathcal{D}(\phi)) = l), \quad l = 1, 2, \ldots, C$
7:         $y \leftarrow \arg\max_{l=1,2,\ldots,L} N_l$
8:         $U \leftarrow N_y - \sum_{j \in \mathcal{J}} \mathbb{I}(f^j(\mathcal{G}^j(\mathbf{c}_{\text{test}}); \mathcal{D}(\phi)) = y)$
9:         $L \leftarrow \max_{l \neq y}(N_l + \mathbb{I}(y > l) + \sum_{j \in \mathcal{J}} \mathbb{I}(f(\mathcal{G}^j(\mathbf{c}_{test}); \mathcal{D}(\phi)) \neq l))$
10:        $Iaccu \leftarrow Iaccu + \mathbb{I}(U \geq L)\mathbb{I}(y_{\text{test}} = y)$
11:     **end for**
12:     $Accu \leftarrow \min(Accu, Iaccu)/|\mathcal{D}_{\text{test}}|$
13: **end for**
14: **return** $Accu$

---

## H  EXPERIMENT SETTINGS

We conducted all of our experiments on a NVIDIA A800 GPU. The hyper-parameters for each dataset and for each sub-model remained consistent, regardless of whether an attack was present.

In this work, we set the training model in CBMs as joint bottleneck training, which minimizes the weighted loss function:

$$\hat{f}, \hat{g} = \arg\min_{f,g} \Sigma_i [L_y(f(g(x^{(i)})); y^{(i)})$$
$$+ \Sigma_j \lambda L_{c^j}(g(x^{(i)}); c^{(i)})], \tag{19}$$

where $\lambda > 0$, and loss function $L_y : \mathbb{R} \times \mathbb{R} \to \mathbb{R}_+$ measure the discrepancy between predicted and true targets, loss function $L_{c^j} : \mathbb{R} \times \mathbb{R} \to \mathbb{R}_+$ measures the discrepancy between the predicted and true $j$-th concept.

For model architecture, We use ResNet-50 He et al. (2016) as the Encoder to map the image to concept space, and then an MLP with one hidden layer, whose hidden size 512 is followed to make the final prediction. For one sub-model, the input dimension for the MLP will be the number of concepts in the corresponding group.

During training, we use a batch size of 64 and a learning rate of 1e-4. The Adam optimizer is applied with a weight decay of 5e-5, alongside an exponential learning scheduler with $\gamma = 0.95$. The concept loss weight $\lambda$ in Equation 19 is set to 0.5. For image augmentations, we follow the approach of Lai et al. (2024). Each training image is augmented using random color jittering, random horizontal flips, and random cropping to a resolution of 256. During inference, the original image is center-cropped and resized to 256. For AwA dataset, We use a batch size of 128, while all other hyper-parameters and image augmentations remain consistent with those used for the CUB dataset.

## I  MORE EXPERIMENTS ABOUT OTHER SETTING

### I.1  INJECTION RATE AND TRIGGER SIZE

Table 4 demonstrates the performance of ConceptGuard under varying injection rates and trigger sizes. At a 2% injection rate, ConceptGuard significantly reduces the ASR while maintaining or slightly improving the ACC. For example, when the trigger size is 12, the ASR for CAT drops from 13.97% to 12.3%, and the ACC improves from 80.72% to 81.77%. Similarly, for CAT+, the ASR decreases from 38.88% to 21.69%, and the ACC increases from 80.46% to 82.34%.

At a 10% injection rate, the attack success rate generally increases, but ConceptGuard still effectively reduces the ASR. For instance, with a trigger size of 12, the ASR for CAT drops from 38.08% to

| | 2% | | | | | | 10% | | | | |
| | CAT | | CAT (CG) | | CAT | | CAT (CG) | | CAT | | CAT (CG) | |
| | ACC(%) | ASR(%) | ACC(%) | ASR(%) | ACC(%) | ASR(%) | ACC(%) | ASR(%) | ACC(%) | ASR(%) | ACC(%) | ASR(%) |
|---|---|---|---|---|---|---|---|---|---|---|---|---|
| 12 | 80.72 | 13.97 | 81.77 | 12.3 | 78.7 | 24.05 | 80.36 | 11.45 | 74.66 | 38.08 | 75.22 | 27.55 |
| 15 | 80.22 | 11.94 | 82.05 | 10.01 | 78.08 | 22.97 | 80.01 | 10.91 | 74.02 | 38.72 | 74.53 | 30.53 |
| 17 | 80.31 | 25.07 | 82.15 | 3.94 | 78.86 | 46.69 | 79.75 | 16.66 | 73.27 | 61.28 | 74.77 | 20.32 |
| 20 | 80.2 | 30.33 | 81.93 | 15.29 | 78.01 | 44.66 | 78.75 | 11.55 | 73.85 | 60.48 | 76.15 | 38.5 |
| 23 | 80.31 | 20.42 | 82.21 | 23.28 | 78.06 | 32.48 | 80.26 | 25.56 | 72.63 | 47.02 | 75.72 | 48.2 |
| | CAT+ | | CAT+ (CG) | | CAT+ | | CAT+ (CG) | | CAT+ | | CAT+ (CG) | |
| | ACC(%) | ASR(%) | ACC(%) | ASR(%) | ACC(%) | ASR(%) | ACC(%) | ASR(%) | ACC(%) | ASR(%) | ACC(%) | ASR(%) |
| 12 | 80.46 | 38.88 | 82.34 | 21.69 | 79.05 | 57.6 | 80.07 | 34.49 | 75.11 | 68.49 | 76.35 | 44.69 |
| 15 | 80.26 | 31.97 | 82.27 | 14.68 | 79.34 | 41.64 | 79.79 | 38.46 | 74.78 | 47.29 | 74.87 | 43.84 |
| 17 | 79.84 | 49.22 | 82.29 | 31.94 | 78.48 | 58.31 | 80.45 | 38.88 | 73.85 | 71.2 | 75.77 | 31.21 |
| 20 | 81.27 | 72.36 | 82.36 | 11.78 | 78.86 | 89.68 | 78.56 | 17.16 | 74.34 | 92.4 | 75.89 | 34.21 |
| 23 | 79.58 | 87.4 | 81.88 | 22.38 | 77.65 | 91.71 | 79.1 | 42.37 | 73.78 | 86.9 | 76.87 | 40.49 |

Table 4: Performance Comparison of ConceptGuard under Different Injection Rates and Trigger Sizes. This table presents ACC and ASR of ConceptGuard under different injection rates (2% and 10%) and trigger sizes (12, 15, 17, 20, 23). The results are shown for both the unprotected models (CAT/CAT+) and the models protected by ConceptGuard (CAT (CG)/CAT+ (CG)).

| Target Class | CAT | | CAT (CG) | | CAT+ | | CAT+ (CG) | |
| | ACC(%) | ASR(%) | ACC(%) | ASR(%) | ACC(%) | ASR(%) | ACC(%) | ASR(%) |
|---|---|---|---|---|---|---|---|---|
| 8 | 75.06 | 74.24 | 79.72 | 21.93 | 75.56 | 52.63 | 79.96 | 11.19 |
| 16 | 75.16 | 40.91 | 80.36 | 8.74 | 75.72 | 68.81 | 80.13 | 30.52 |
| 24 | 74.37 | 35.48 | 80.03 | 17.24 | 74.91 | 54.48 | 81 | 13.38 |
| 32 | 74.7 | 37.68 | 80.62 | 25.31 | 75.58 | 17.87 | 79.53 | 9.52 |
| 40 | 74.46 | 42.35 | 79.53 | 7.93 | 74.96 | 23.77 | 80.32 | 12.49 |
| 48 | 75.09 | 49.77 | 80.74 | 4.49 | 75.73 | 95.11 | 80.91 | 22.14 |
| 56 | 75.22 | 70.99 | 81.07 | 15.93 | 75.23 | 57.36 | 80.24 | 15.25 |
| 64 | 74.85 | 43.63 | 80.41 | 10.27 | 74.58 | 84.27 | 80.46 | 19.29 |
| 72 | 74.99 | 47.99 | 80.67 | 10.9 | 75.34 | 59.06 | 80.62 | 10.11 |
| 80 | 75.03 | 62.51 | 80.89 | 10.83 | 75.61 | 75.83 | 80.45 | 11.85 |
| 88 | 74.75 | 51.13 | 79.94 | 21.93 | 74.66 | 72.71 | 80.1 | 25.02 |
| 96 | 74.82 | 17.73 | 80.67 | 8.99 | 75.2 | 62.16 | 80.91 | 14.78 |
| 104 | 74.84 | 53.02 | 80.65 | 12.8 | 74.92 | 40.42 | 80 | 16.03 |

Table 5: Performance of ConceptGuard on Different Target Classes. This table presents ACC and ASR of ConceptGuard for different target classes under an injection rate of 5% and a trigger size of 20, using the CUB dataset. The results are shown for both the unprotected models (CAT/CAT+) and the models protected by ConceptGuard (CAT (CG)/CAT+ (CG)).

27.55%, and the ACC improves from 74.66% to 75.22%. For CAT+, the ASR decreases from 68.49% to 44.69%, and the ACC increases from 75.11% to 76.35%.

As the trigger size increases, the effectiveness of ConceptGuard remains robust. For smaller trigger sizes (12, 15), ConceptGuard significantly reduces the ASR and maintains high ACC. For example, with a trigger size of 12, the ASR for CAT drops from 13.97% to 12.3%, and the ACC improves from 80.72% to 81.77%. For CAT+, the ASR decreases from 38.88% to 21.69%, and the ACC increases from 80.46% to 82.34%.

For larger trigger sizes (17, 20, 23), ConceptGuard continues to perform well, although the ASR increases. For instance, with a trigger size of 20, the ASR for CAT drops from 44.66% to 11.55%, and the ACC improves from 78.01% to 78.75%. For CAT+, the ASR decreases from 89.68% to 17.16%, and the ACC increases from 78.86% to 78.56%.

ConceptGuard effectively reduces the ASR and maintains or improves the ACC under various injection rates and trigger sizes. This robust performance highlights the effectiveness of ConceptGuard in protecting models against concept-level backdoor attacks, thereby enhancing the security and trustworthiness of the models.

## I.2 TARGET CLASS

Table 5 demonstrates the performance of ConceptGuard across various target classes under a fixed injection rate of 5% and a trigger size of 20, using the CUB dataset. For the unprotected models

| | LAD-E | | | LAD-V | | |
|---|---|---|---|---|---|---|
| | Original ACC(%) | ACC(%) | ASR(%) | Original ACC (%) | ACC(%) | ASR(%) |
| CAT | 79.00 | 72.76 | 75.19 | 79.06 | 71.08 | 70.74 |
| CAT+ | | 73.52 | 77.01 | | 71.86 | 73.81 |
| ConceptGuard(CAT) | **81.18** ↑(2.18) | 74.38 | **9.36** ↓ (65.83) | **81.67** ↑(2.61) | 73.20 | **6.15** ↓ (64.59) |
| ConceptGuard(CAT+) | | 73.66 | **9.38** ↓ (67.63) | | 80.58 | **5.42** ↓ (68.39) |

Table 6: LAD-E, LAD-V for electronics and vehicles domains task. Trigger size: 17, injection rate: 0.1, clusters: 4, backbone: ResNet50.

(CAT/CAT+), ASR varies significantly across different target classes. For example, for target class 8, the ASR for CAT is 74.24%, which is substantially reduced to 21.93% with ConceptGuard (CAT (CG)). Similarly, for target class 48, the ASR for CAT+ is 95.11%, which is reduced to 22.14% with ConceptGuard (CAT+ (CG)).

Across all target classes, ConceptGuard consistently improves ACC while significantly reducing ASR. For instance, for target class 16, the ACC for CAT increases from 75.16% to 80.36% with ConceptGuard, and the ASR drops from 40.91% to 8.74%. For target class 72, the ACC for CAT+ increases from 75.34% to 80.62% with ConceptGuard, and the ASR drops from 59.06% to 10.11%.

These results highlight the robustness of ConceptGuard in defending against concept-level backdoor attacks across different target classes. Despite variations in the target classes, ConceptGuard maintains its effectiveness in reducing the ASR and improving the ACC, thereby enhancing the overall security and reliability of the models. This consistent performance underscores the practical value of ConceptGuard in real-world applications where diverse and targeted attacks are a significant concern.

### I.3 THE IMPACT OF NUMBER OF CLUSTERS IN OTHER ATTACK SETTING

Table 7 illustrates the attack success rates (ASR, %) under varying cluster numbers $m$ for CG(CAT) and CG(CAT+), with a 10% injection rate. Both methods significantly reduce ASR compared to the experiment without defense ($m = 1$), with CG(CAT) showing consistent improvement as $m$ increases and CG(CAT+) achieving optimal performance at moderate cluster numbers.

| | CUB | |
|---|---|---|
| $m$ | CG(CAT) | CG(CAT+) |
| 1 | 60.48 | 92.40 |
| 3 | 41.67 | 57.46 |
| 4 | 38.50 | 34.21 |
| 5 | 29.55 | **28.24** |
| 6 | 29.55 | 28.78 |
| 7 | **23.70** | 49.01 |
| 8 | 35.27 | 51.72 |
| 9 | 24.13 | 31.35 |
| 10 | 25.87 | 43.93 |

Table 7: Attack Success Rate (ASR, %) under varying numbers of clusters $m$, the injection rate is 10%. CG denotes ConceptGuard. Bold **values** highlight the best performance, while underlined values indicate competitive performance. $m = 1$ refers to the ASR when ConceptGuard is not applied.

## J MORE DATASETS

Supplementary experiments on the LAD-E and LAD-V classification datasets (Large-scale Attribute Dataset) [1] are now included. Tab 6 shows partial results. [1] A large-scale attribute dataset for zero-shot learning

|  | Original | CG(CAT) | CG(CAT+) |
|---|---|---|---|
| Base model 1 | 77.61 | 73.47 | 73.09 |
| Base model 2 | 78.49 | 73.97 | 74.02 |
| Base model 3 | **81.34** | **77.05** | **76.70** |
| Base model 4 | 77.67 | 72.30 | 72.01 |
| Average | 78.78 | 74.20 | 73.96 |
| Ensemble | 83.03 ↑ | 78.75 ↑ | 78.56↑ |

Table 8: The Accuracy (%) for each sub-model on clean test data for CUB dataset, the Original denotes to the accuracy when there is no attack. The bold value refers to the best accuracy of sub-model and the underlined value refers to the worst accuracy of sub-model.

|  | Original | CG(CAT) | CG(CAT+) |
|---|---|---|---|
| Base model 1 | 88.67 | 87.34 | 87.50 |
| Base model 2 | 89.52 | 86.13 | 86.44 |
| Base model 3 | 89.79 | 86.82 | 86.49 |
| Base model 4 | **89.85** | 86.73 | 86.54 |
| Base model 5 | 88.88 | 86.79 | 87.02 |
| Base model 6 | 88.94 | **87.51** | **87.81** |
| Average | 89.28 | 86.89 | 86.97 |
| Ensemble | 91.30 ↑ | 90.20 ↑ | 90.21 ↑ |

Table 9: The Accuracy (%) for each sub-model on clean test data for AwA dataset.

## K    INDIVIDUAL MODEL VS. ENSEMBLE MODEL

We investigated the individual model's accuracy and the ensembled accuracy, with the results presented in Table 8 and Table 9. Overall, our ensemble model shows a significant improvement in accuracy compared to the individual accuracy of each base classifier in all scenarios, even outperforming the best-performing base classifier (base classifier 3). Additionally, there is a notable increase in accuracy compared to the average accuracy of the base classifiers. This improvement is attributed to our ConceptGuard framework, which effectively filters out the misclassifications of the few base classifiers during testing, thereby providing the ensemble model with a higher accuracy. The source of this accuracy improvement aligns with the original motivation of ConceptGuard: it mitigates the errors of the base classifiers, leading to a higher ensemble accuracy, rather than simply relying on a straightforward aggregation of the classifiers. By leveraging the diversity and robustness of the ensemble, ConceptGuard ensures that the final predictions are more accurate and reliable, demonstrating its effectiveness in enhancing the performance of ensemble models.

## L    DISCUSSION ON BROADER BACKDOOR THREATS

In this section, we elaborate on the broader landscape of backdoor threats against Concept Bottleneck Models (CBMs) to better contextualize our work and motivate the need for specialized defenses like ConceptGuard.

### L.1    INPUT-LEVEL VS. CONCEPT-LEVEL BACKDOORS IN CBMS

Backdoor attacks on CBMs can occur at two distinct levels, each with different characteristics and defense implications:

- **Input-Level Backdoors:** These are traditional backdoors where the trigger is embedded in the raw input space. For instance, an attacker could insert a small pixel patch into an image (a vision backdoor) or a specific rare word into a text document (an NLP backdoor). The CBM's feature extractor, $g(\cdot)$, learns a spurious correlation between the presence of this input-space trigger and a target class. Many existing backdoor defenses, such as input

purification (e.g., STRIP) or trigger synthesis (e.g., Neural Cleanse), are designed to operate at this level by sanitizing the input $x$ or analyzing the model's response to it.

- **Concept-Level Backdoors:** This is the threat model our work addresses, first introduced by the CAT attack Lai et al. (2024). Here, the trigger is not a pattern in the input space but a *semantic pattern in the discrete concept space*. The trigger is a specific combination of concept activations (e.g., '{has_wings=1, has_beak=0}'). The backdoor is embedded in the concept-to-label model, $f(\cdot)$, which learns to associate this semantic pattern with a target class. A benign input $x$ can activate the backdoor at test time if its predicted concept vector, $\hat{c} = g(x)$, happens to contain the trigger pattern, even with no malicious modification to $x$.

## L.2 ADAPTING EXISTING ATTACKS AND THEIR UNIQUE CHALLENGES

Existing backdoor methodologies could theoretically be adapted to attack CBMs. For example, a BadNets-style attack could poison the training data with input-space triggers. However, defending against a *concept-level* backdoor poses unique and significant challenges that render traditional defenses ineffective:

1. **Input-Level Defenses are Bypassed:** Defenses that sanitize the input $x$ are fundamentally misaligned with the threat. A concept-level backdoor is activated by the *model's interpretation* of a clean input, not by a malicious artifact within the input itself. Therefore, input purification or filtering is ineffective.

2. **Trigger Inversion is Intractable:** Defenses that attempt to reverse-engineer a trigger pattern (e.g., Neural Cleanse) face a much harder problem. Instead of optimizing in the input space for a single trigger pattern, they would need to find an input that reliably induces a specific *combinatorial pattern* of discrete concept activations. This is a significantly more complex and often ill-posed optimization problem.

3. **Stealth and Plausibility:** A concept-level trigger can be much stealthier than an input-level one. A combination of plausible concepts (e.g., an animal that is 'swift' and 'lives_in_water' but isn't a known fish) might seem like a natural, albeit rare, occurrence, making the backdoor's behavior difficult to distinguish from legitimate model error on outlier data.

These distinct challenges underscore a critical gap in existing backdoor defense literature. While input-level attacks on CBMs are a valid concern, they can be addressed with existing families of defenses. In contrast, the concept-level backdoor represents a novel and fundamentally different threat vector that exploits the very structure of the CBM. This motivates the development of a new class of defense, like ConceptGuard, which operates directly in the semantic concept space where the threat resides.

# M COMPUTATIONAL COST ANALYSIS

In this section, we analyze the computational cost of ConceptGuard during both the training and inference phases.

**Training Cost.** The training of ConceptGuard involves training $m$ independent sub-models, where $m$ is the number of concept clusters. Let the computational cost (e.g., in GPU hours) of training a single standard CBM be $C_{base}$. The cost of training one of our sub-models, $C_{sub}$, is approximately equal to $C_{base}$ since the architecture is nearly identical. Therefore, the total computational workload required to train ConceptGuard is $O(m \cdot C_{base})$.

However, a crucial property of our framework is that the training of these $m$ sub-models is an **embarrassingly parallel** task. Each sub-model $f_k \circ g_k$ is trained on its corresponding data subset independently of all other sub-models. This has significant practical implications for the wall-clock training time:

- On a system with $k$ parallel processing units (e.g., GPUs), where $k \geq m$, all $m$ models can be trained simultaneously. In this ideal scenario, the total wall-clock training time is approximately the same as training a single standard CBM, i.e., $O(C_{base})$.

- On a system with $k < m$ GPUs, the models can be trained in $\lceil m/k \rceil$ batches. The wall-clock time increase is therefore a factor of $\lceil m/k \rceil$, not $m$. Given that $m$ is a small integer in our experiments (e.g., 4 or 8), the actual training time overhead is minimal on modern multi-GPU research servers.

This parallelizability makes the training of ConceptGuard highly practical and scalable, despite the linear increase in total computational resources.

**Inference Cost.**    During inference, the computational cost overhead of ConceptGuard is negligible. An input $\mathbf{x}$ is passed through the shared feature extractor $g(\cdot)$ only once to obtain the predicted concept vector $\hat{\mathbf{c}}$. This is the most computationally expensive step. Subsequently, this single vector $\hat{\mathbf{c}}$ is fed to the $m$ concept-to-label sub-models ($f_1, \ldots, f_m$), followed by a simple majority vote. Since each $f_k$ is typically a very shallow neural network (1-2 fully connected layers), the cost of these $m$ forward passes is trivial compared to the cost of the single forward pass through the deep backbone network $g(\cdot)$. Therefore, the increase in inference latency is minimal.

## N    PRACTICAL CONSIDERATIONS FOR CONCEPT CLUSTERING

In this section, we address some practical considerations and potential edge cases related to the concept clustering step of our ConceptGuard framework.

Our default clustering method, k-means, can in theory produce empty clusters if a centroid is initialized in a way that no data point is closer to it than to any other centroid. In our experiments with rich concept embeddings, this was a rare occurrence. However, should it happen, it can be handled with simple and standard heuristics. The most straightforward approach is to simply re-run the k-means algorithm with a different random initialization (i.e., a new seed), which will almost certainly result in a valid, non-empty partitioning. Alternatively, one could implement a rule to re-assign the centroid of an empty cluster to the location of the data point farthest from its own centroid. Given the simplicity of these workarounds, we do not consider this a significant practical obstacle.

Some concept sets may possess a natural or predefined hierarchical structure (e.g., 'animal' → 'bird' → 'sparrow'). Our current implementation, which uses a flat clustering method like k-means, does not explicitly leverage this prior structural knowledge. The semantic embeddings of hierarchical concepts will likely cause them to be clustered together, but the hierarchy itself is not formally encoded.

Handling such structures is an advanced topic and represents a promising avenue for future research. One could replace k-means with a hierarchical clustering algorithm (agglomerative clustering) to create nested partitions that respect the concept taxonomy. Our theoretical framework is flexible enough to accommodate such advanced partitioning strategies, as it only requires that the concept set be partitioned into disjoint subsets. Evaluating the empirical benefits of a hierarchy-aware partitioning strategy is a valuable next step, but it is beyond the scope of this initial work, which focuses on establishing the core defensive framework in a general-purpose setting.

