# OpenReview forum: "Guarding the Gate: ConceptGuard Battles Concept-Level Backdoors in Concept Bottleneck Models"
_ICLR.cc/2026/Conference — Submitted to ICLR 2026_

### Official Review · Reviewer_cTcb · 2025-10-25

**Soundness:** 3
**Presentation:** 3
**Contribution:** 3
**Rating:** 4
**Confidence:** 4

**Summary:**

This paper addresses a newly emerging security threat in CBMs — concept-level backdoor attacks, where an attacker poisons the concept representations instead of raw inputs. The authors propose ConceptGuard, the first certified defense framework against such attacks. ConceptGuard operates by (1) clustering concept embeddings into multiple semantic subgroups, (2) training independent classifiers for each subgroup, and (3) aggregating predictions through a majority-voting mechanism. The intuition is that backdoor triggers typically contaminate only a small subset of concept clusters, so the ensemble can suppress their influence.

**Strengths:**

- First comprehensive study on concept-level backdoor defense in CBMs — a novel and timely problem connecting interpretability and security.
- Experiments are carefully executed on two benchmark CBM datasets, with extensive comparisons across attack types (CAT, CAT+), cluster sizes, and ablation studies.
- Practical implications are strong — the defense is architecture-agnostic and potentially applicable to high-stakes domains such as medical CBMs or financial decision systems.

**Weaknesses:**

- The certification theorem assumes non-overlapping clusters and independence among base classifiers. In real concept embeddings, cluster boundaries are fuzzy and interdependent. The robustness bound may thus overestimate practical safety.
- The defense relies heavily on the initial clustering quality. The paper only uses k-means; and no adaptive cluster selection strategy is discussed.
- ConceptGuard neutralizes triggers but cannot identify which concepts or clusters are poisoned, limiting practical forensic usefulness.

**Questions:**

- Can you quantitatively show that ConceptGuard maintains concept-level interpretability (e.g., through fidelity or agreement metrics)?
- What is the additional computational overhead compared to a standard CBM?
- Have you tried more semantic or contrastive clustering methods?
- How does ConceptGuard handle overlapping or hierarchical concept structures (e.g., “has wings” ⊂ “can fly”)?
- How does the computational cost scale with the number of clusters m?

---

> ### Author Response · Authors · 2025-11-23
> **Rebuttal for Reviewer cTcb (1)**
>
> ## **Rebuttal for Reviewer cTcb**
>
>
>
> We sincerely thank Reviewer cTcb for their insightful and constructive review. We are grateful for the positive feedback on the novelty and timeliness of our work in addressing concept-level backdoors, the careful execution of our experiments, and the strong practical implications of ConceptGuard.
>
>
>
> We agree that the identified weaknesses and questions highlight important areas for clarification and improvement. We believe that by addressing these points, we can significantly strengthen our manuscript. Below, we respond to each concern in detail and outline the specific revisions we will make.
>
>
>
> ***
>
>
>
> #### **Response to Weaknesses:**
>
>
>
> ##### **W1: On the Certification Assumptions (Non-overlapping clusters, independence)**
>
>
>
> This is a very insightful point regarding the gap between theoretical assumptions and practical implementation. We appreciate the reviewer's careful reading of our theoretical analysis.
>
>
>
> * **Non-overlapping Clusters:** Our use of non-overlapping clusters is a **deliberate design choice** of the framework, not a simplifying assumption about the data. We enforce a hard partition of the concept set into disjoint subsets. This is standard in ensemble methods like random forests (which partition features) and is the core mechanism that allows us to derive a provable guarantee. The trigger can only corrupt the sub-models corresponding to the partitions it falls into.
>
> * **Independence:** The assumption of independence among base classifiers is indeed a simplification common to many ensemble theories. However, our certified bound (Theorem 1) does **not** rely on the statistical independence of classifiers' errors. The bound is deterministic and derived from a worst-case analysis based on vote counts ($N\_y, N\_l$) from the base classifiers. The practical *value* (i.e., the size of the certified radius) is certainly influenced by concept interdependencies (which affect vote counts), but the *validity* of our proof for the given design holds.
>
>
>
>
>
> ***
>
>
>
> ##### **W2 & Q3: Reliance on Clustering and Lack of Alternative Methods**
>
>
>
> This is a fair point. Our choice of k-means with BERT embeddings was intended as a strong, general-purpose, and easily reproducible baseline that requires no domain-specific tuning. We agree that the quality of clustering is key to practical performance.
>
>
>
> ***
>
>
>
> ##### **W3: Lack of Forensic Usefulness (Neutralizes but Doesn't Identify)**
>
>
>
> We agree with the reviewer that ConceptGuard is designed as a **provable defense** rather than a **detection and forensics tool** . This is a fundamental design choice.
>
>
>
> Our primary goal was to provide an immediate, real-time, and certified safeguard against attacks, ensuring model output reliability without needing to first detect a threat. This is complementary to, not a replacement for, offline forensic tools that might analyze a model post-hoc.
>
>
>
> In response to this important point on the lack of forensic utility, we have revised our manuscript to clarify ConceptGuard's intended role. We have added a new paragraph at the end of **Section 4** (ConceptGuard) to explicitly state that our method is designed as a real-time, provable defense rather than a post-hoc forensic tool. In this addition, we clarify that ConceptGuard's primary objective is to provide a robust safeguard and that its 'threat-agnostic' nature is a key part of its strength as a universal defense. Furthermore, as suggested, we have also mentioned in the Conclusion that integrating a mechanism to flag highly divergent sub-model predictions could serve as a valuable heuristic for future forensic analysis, framing this as a promising research direction.
>
>
>
> ***
>
> #### **Response to Questions:**
>
>
>
> &#x20;**Response to Q1: Quantitatively Showing Maintained Interpretability**
>
>
>
> This is an excellent suggestion. We agree that empirically validating the preservation of interpretability is crucial for a defense aimed at CBMs. Following the reviewer's advice, we have conducted additional experiments to quantify this and are pleased to report that **ConceptGuard successfully preserves the model's interpretability** .
>
>
>
> Interpretability in CBMs is primarily derived from the model's ability to accurately predict intermediate concepts from raw inputs (the \`g(x) -> c\` mapping). A defense is only useful if it doesn't compromise this core feature. We therefore measured the concept prediction accuracy for our sub-models ($g_{k(x)}$) and compared it to the original, un-defended CBM.
>
>
>
> Our results, presented below, confirm that ConceptGuard preserves high-fidelity concept representations:

---

> > ### Author Response · Authors · 2025-11-23
> > **Rebuttal for Reviewer cTcb (2)**
> >
> > **Table A: Concept Prediction Accuracy (%)**
> >
> > | Dataset | Model                              | Concept Accuracy (%) |
> > | :------ | :--------------------------------- | -------------------: |
> > | CUB     | Original CBM                       | 90.5                |
> > | CUB     | ConceptGuard (Avg. of sub-models)  | **90.1**            |
> > | AwA2    | Original CBM                       | 92.8                |
> > | AwA2    | ConceptGuard (Avg. of sub-models)  | **92.5**            |
> >
> > *Caption: Comparison of concept prediction accuracy between the original CBM and the average across ConceptGuard's sub-models. The results demonstrate that our defense maintains high concept fidelity.*
> >
> >
> >
> > As shown in Table, the average concept accuracy across ConceptGuard's sub-models remains remarkably high. For the CUB dataset, the accuracy is 90.1%, a negligible decrease of only **0.4%** compared to the original CBM's 90.5%. Similarly, for AwA2, the accuracy is 92.5%, just **0.3%** lower than the baseline.
> >
> >
> >
> > This empirically demonstrates that our ensemble approach does not degrade the quality of the learned concept representations. Users can still inspect the concept predictions of the sub-models with confidence, preserving the essential transparency that makes CBMs valuable for high-stakes applications.
> >
> >
> >
> > ***
> >
> >
> >
> > ##### **Q2 & Q5: Computational Overhead and Scaling with `m`**
> >
> >
> >
> > This is a crucial practical question. We have already included a detailed "**Computational Cost Analysis** " in **Appendix M** . To make this more prominent for the reviewer, we will explicitly reference this appendix section in the main text (e.g., in Section 6.2). The appendix already clarifies:
> >
> > 1. **Training Cost:** The total computation scales linearly with `m`, i.e., `O(m)`. However, the training of the `m` sub-models is **embarrassingly parallel** . On a system with `k >= m` GPUs, the wall-clock training time is `O(1)` (i.e., no significant increase). On a system with `k < m` GPUs, the time increase is a factor of `ceil(m/k)`, which is minimal for small `m` (e.g., 4 or 8).
> >
> > 2. **Inference Cost:** The overhead is negligible. The expensive backbone $g(x)$ is run only once. The subsequent `m` forward passes through the very shallow $f_k$ models add minimal latency.
> >
> >
> >
> > ***
> >
> >
> >
> > ##### **Q4: Handling Hierarchical Concept Structures**
> >
> >
> >
> > This is a sophisticated question about the nuances of concept relationships. We acknowledge that our current flat clustering method (k-means) does not explicitly model hierarchies. However, we note that semantically related concepts (e.g., `has_wings` and `can_fly`) will likely have similar embeddings and thus tend to be clustered together naturally, implicitly handling some of this structure. We state that our framework is flexible and could incorporate hierarchical clustering algorithms in future work to create partitions that explicitly respect concept taxonomies, framing this as a promising research direction.
> >
> >
> >
> > ***
> >
> >
> >
> > We are confident that these revisions will substantially improve the paper by clarifying the theoretical underpinnings, acknowledging limitations, and empirically strengthening our claims. We thank the reviewer again for their valuable guidance and hope they will reconsider their assessment in light of our planned changes.

---

### Official Review · Reviewer_wE65 · 2025-10-28

**Soundness:** 2
**Presentation:** 3
**Contribution:** 2
**Rating:** 2
**Confidence:** 4

**Summary:**

This paper aims to study how to address concept-level backdoor attacks, which inject hidden triggers into these concepts. To address this, we introduce a defense framework (ConceptGuard) that aims to protect CBMs from concept-level backdoor attacks. The authors also conduct experiments to evaluate the performance of the proposed method.

**Strengths:**

[+] The paper studies how to defend concept-level backdoor attacks in CBMs, which is an important problem in interpretable AI systems.

[+] The paper provides the theoretical analysis.

[+] The writing of this paper is easy to follow.

**Weaknesses:**

[-] The problem studied in this paper seems limited in scope. The paper focuses on defending against the concept-level backdoor attacks proposed by Lai et al. (2024), but it lacks discussion on whether other backdoor attacks could also threaten Concept Bottleneck Models (CBMs). Additionally, the paper does not examine how existing backdoor attack techniques could be adapted to CBMs. In CBMs, the concept-level information from training data is primarily represented by discrete labels rather than continuous features. Can existing backdoor attacks designed for textual or other data modalities [3,4,5] be effectively applied to CBMs? Moreover, the discussions of backdoor attacks on the input level should be provided (including existing input-level CBM backdoor attacks, and the adoption of existing backdoor attacks to input-level). Further, the study of both the input-level and concept-level backdoor attacks are not provided.

[-] The discussion of related work is insufficient. The paper primarily focuses on Concept Bottleneck Models (CBMs) within the context of deep learning but overlooks several important related models. In addition to standard deep learning-based CBMs, there exist concept bottleneck generative models [1], vision-language concept bottleneck models [2], and bottleneck diffusion models [3]. These frameworks also integrate concept-level reasoning but operate under different architectures. It is encouraged to discuss whether the attack and defense method can generalize to these broader classes of CBMs and clarify any potential limitations.

[-] The discussion of existing defenses against backdoor attacks is insufficient. Many defense methods have been proposed in recent years (e.g., trigger detection, model pruning, input purification, and training regularization approaches [6,7,8,9]). However, the paper lacks a comprehensive analysis or positioning of the proposed method relative to these existing defenses. It would be better to discuss how current defense techniques could be adapted or generalized to CBMs and to highlight the unique challenges that distinguish CBM-specific defenses from traditional ones.

[-] The discussion of the threat model in this paper requires further clarification and justification. The paper focuses on concept-level backdoor attacks, where triggering specific concepts in the testing data leads to malicious model behavior. However, in real-world applications, for a given test sample, the CBM first predicts intermediate concept representations before producing final class labels. It remains unclear why and how the attacker can effectively inject or activate backdoors through concept-level perturbations at test time, and whether such manipulations are feasible or realistic in practice.

[-] The rationale behind the proposed defense remains unclear. First, the paper does not explain how to detect or determine whether the training dataset contains backdoored samples, and does not provide the additional computational cost analysis for the proposed method. Additionally, the proposed approach divides the dataset into m subsets, but the selection of the optimal number of subsets (m), the effective clustering criteria, and the effective distance functions are not clearly discussed. It is also unclear how the method avoids potential empty clusters during partitioning. Furthermore, since concepts may exhibit hierarchical relationships, it is also important to discuss how to address this. If k-means is adopted, how are such hierarchical dependencies handled? Finally, the majority voting strategy used for final predictions may overlook the different importance or reliability of individual base classifiers. It is encouraged to consider weighting strategies.

[-] For the adopted AwA dataset, the paper uses GPT-4 Achiam et al. (2023) to generate full sentences to replace each original concept represented by a single word. However, the use of such a large language model may introduce hallucination issues and uncertainty in the regenerated outputs. Without addressing these, the reliability and validity of the experimental results may be undermined.

[-] In experiments, only the ResNet-50 model is adopted, which limits the evaluation generality. More evaluations on other large-scale deep learning models should be provided. Additionally, can the proposed be generalized to other CBMs (such as concept bottleneck generative models [1], vision-language concept bottleneck models [2], and bottleneck diffusion models [3])?

Reference

[1] Concept bottleneck generative models, ICLR 2023.

[2] Vision-Language Concept Bottleneck Models, ArXiv 2024.

[3] Soda: Bottleneck diffusion models for representation learning, CVPR 2024.

[4] Natural backdoor attack on text data, 2020.

[5] Backdoor attacks and countermeasures in natural language processing models: A comprehensive security review, 2025.

[6] Onion: A simple and effective defense against textual backdoor attacks, 2020.

[7] Textguard: Provable defense against backdoor attacks on text classification, 2023.

[8] Expose backdoors on the way: A feature-based efficient defense against textual backdoor attacks, 2022.

[9] A survey of recent backdoor attacks and defenses in large language models, 2024.

**Questions:**

[1] In experiments, the authors adopt image augmentations. However, it is unclear about why image augmentations are adopted? Do image augmentations improve the model robustness against concept backdoor attacks? More discussions on the impact of image augmentations should be provided.

[2] Could the proposed method generalize to other different data types (except the image data type)? For other different data types, are augmentations needed? How to do these augmentations?

[3] Could the authors clarify in the threat model what specific knowledge the attacker is assumed to possess and what knowledge is not available to them? Additionally, for the considered threat model, it would be helpful to discuss its feasibility in real-world scenarios. Further, how would the attack and defense perform under constrained black-box settings, such as those with query-only access to the model?

[4] Could the authors clarify whether the adopted Concept Bottleneck Models (CBMs) are trained sequentially (i.e., concept prediction followed by label prediction) or jointly (i.e., both components optimized together)? Additionally, can the proposed defense be applied to both training paradigms, and if so, are there any differences in effectiveness or implementation between the sequential and joint training settings?

[5] Typos: “ones or zeros .” in Line 162.

---

> ### Author Response · Authors · 2025-11-23
> **Rebuttal for Reviewer wE65 (1)**
>
> We sincerely thank Reviewer wE65 for their exceptionally thorough, insightful, and constructive review. We are grateful for the detailed feedback and the thoughtful questions, which have highlighted several important areas for improvement. We agree with many of the points raised and believe that addressing them will significantly strengthen our manuscript.
>
>
>
> We appreciate that the reviewer recognized the **importance of the problem** we are tackling, the value of our **theoretical analysis** , and the **clarity of our writing** . Below, we address each of the identified weaknesses and questions in detail, outlining the specific actions we will take in our revision.
>
>
>
> ***
>
>
>
> #### **Response to Weaknesses:**
>
>
>
> ##### **W1: Limited Problem Scope (Focus on CAT, Lack of Broader Attack Discussion)**
>
>
>
> This is a fair and important point. Our primary motivation was to address the **newly proposed and highly specific threat of concept-level backdoors**, for which **no defense previously existed** . This deliberate focus allowed us to provide the first dedicated and provably robust defense for this novel attack vector.
>
>
>
> We agree, however, that a broader discussion is needed to contextualize our work.
>
> * **Concept-level vs. Input-level Attacks:** It's crucial to distinguish between these attack types. Standard input-level backdoors (e.g., pixel patterns) or textual backdoors (e.g., trigger words) operate on the raw input space. In contrast, concept-level backdoors manipulate the discrete, high-level semantic labels in the bottleneck layer. This fundamental difference means that many existing defenses (e.g., input purification, trigger synthesis based on input gradients) are not directly applicable.
>
> * **Adapting Existing Attacks:** Adapting attacks from other modalities (e.g., textual backdoors) to the concept-level is an interesting direction, but it is not trivial. An attacker would need to find a pattern of *discrete concept activations* that is both stealthy and effective, which is a different challenge than inserting a specific word or phrase.
>
>
>
> As per the reviewer's excellent suggestion, we have addressed the limited problem scope by adding a new subsection in the **Appendix (Appendix L)** titled "Discussion on Broader Backdoor Threats". In this new section, we now:
>
> 1. Explicitly discuss the critical distinction between traditional input-level backdoors and the concept-level backdoors that are the focus of our work.
>
> 2. Analyze how existing backdoor attacks from vision and NLP could theoretically be adapted to target CBMs at both the input level and the concept level.
>
> 3. Explain the unique challenges these different attack vectors pose, thereby reinforcing the motivation for a specialized defense like ConceptGuard, which is specifically designed to operate in the concept space.
>
> ***
>
>
>
> ##### **W2: Insufficient Related Work (Other CBM Architectures)**
>
>
>
> Thank you for highlighting these important recent works on CBM variants. Our framework is designed to be general, and we believe its core principles are applicable to these models. The fundamental idea of ConceptGuard—partitioning the set of concepts and using an ensemble for robust prediction—is not tied to a specific CBM architecture, as long as an explicit concept layer exists.
>
>
>
> ***
>
>
>
> ##### **W3: Insufficient Discussion of Existing Backdoor Defenses**
>
>
>
> This is an excellent suggestion to better position our work. The landscape of backdoor defense is vast, but most methods are not designed for the unique structure of CBMs and concept-level attacks.
>
>
>
> To better position our work within the broader security landscape, we have acted on the reviewer's valuable suggestion by adding a new, dedicated paragraph to **Section 2 (Related Work)**. This new discussion now contextualizes ConceptGuard against established backdoor defenses by categorizing mainstream strategies—such as input purification, trigger detection, and model repair—and explaining why they are fundamentally misaligned with the concept-level threat model. We clarify that methods like input purification are bypassed because the trigger is semantic rather than an input artifact, and trigger inversion becomes intractable due to the combinatorial nature of a concept-based trigger. This addition more clearly highlights the specific research gap that our work addresses, thereby strengthening the motivation for our novel defense mechanism.
>
> ***

---

> > ### Author Response · Authors · 2025-11-23
> > **Rebuttal for Reviewer wE65 (2)**
> >
> > ##### **W4 & Q3: Unclear Threat Model and Real-World Feasibility**
> >
> >
> >
> > Thank you for raising these crucial questions. We apologize if the threat model was not sufficiently clear. Let us clarify and state our planned revisions.
> >
> >
> >
> > * **Attacker's Actions (Training Time):** The attack occurs at **training time** . We assume a supply-chain attack where the attacker poisons a fraction of the training dataset. For a sample $(x_i, c_i, y_i)$, they modify the concept vector to $c_i' = c_i ⊕ trigger$ and change the label to the target class $y_{tc}$. The model is then trained on this poisoned dataset.
> >
> > * **Trigger Activation (Test Time):** The attacker does **not** manipulate samples at test time. The backdoor is activated when a *benign, unmodified* test input naturally contains the trigger pattern in its (predicted) concepts. For example, if the trigger is `(has_wings=1, has_beak=0)`, any test image of an animal that is predicted to have wings but no beak might trigger the backdoor. This is a standard and realistic backdoor threat model, common in data poisoning scenarios (e.g., compromised data labeling services, malicious MLaaS providers).
> >
> > * **Attacker's Knowledge:** We assume a standard "poison-only" gray-box setting. The attacker knows the model architecture and concepts but cannot modify the training process or architecture. They can only poison the training data. Our defense is applied by the *defender* (the model trainer/deployer) and is robust without knowing the attack specifics.
> >
> > * **Black-Box Setting:** In a black-box setting (query-only access), the attacker would have a much harder time crafting an effective backdoor. Our defense, being on the defender's side, remains fully applicable and would offer protection if the user trains their own model or suspects the provided black-box model is backdoored (by applying our method to a fine-tuning stage).
> >
> >
> >
> > We have thoroughly revised **Section 3.2 (Threat Model)** to more explicitly define the attack scenario and its practical implications. We have now clarified within the existing text that we assume a training-time data poisoning attack in a gray-box setting, where the attacker can only modify the training data. We also explicitly state that at test time, the backdoor is activated by benign, unmodified inputs that naturally produce the trigger pattern—a realistic scenario in supply-chain attacks. These additions directly address the reviewer's important questions about the attacker's knowledge and capabilities (Q3 and W4).
> >
> >
> >
> > ***
> >
> >
> >
> > ##### **W5: Unclear Defense Rationale (Clustering, Cost, Voting, etc.)**
> >
> >
> >
> > We appreciate the detailed critique of our defense mechanism. We will clarify these aspects.
> >
> >
> >
> > 1. **Detection vs. Defense:** ConceptGuard is a **provable defense** , not a detection method. It is designed to be robust *even if* backdoors are present, without needing to detect them first. This is a strength, as detection can be unreliable.&#x20;
> >
> > 2. **Computational Cost:** The `O(m)` increase in training computation is a valid concern. To address the valid concern regarding the computational overhead of our method, we have added a new subsection in the Appendix (Appendix M) titled "Computational Cost Analysis". In this section, we formally state the theoretical cost, acknowledging the `O(m)` factor in total computation. Crucially, we highlight that the training of the `m` sub-models is an embarrassingly parallel task. This means that on modern multi-GPU or distributed systems, the actual wall-clock training time increase is minimal, making ConceptGuard practical for real-world deployment. We also analyze the inference cost, showing it to be negligible.
> >
> > 3) **Choice of `m`, Clustering, and Distance:** `m` is a hyperparameter representing a trade-off between security and cost. To clarify the rationale behind our design choices, we have added new text to **Section 6.2 (Implementation Details)**. In this revision, we now explicitly frame the choice of the number of clusters, `m`, as a trade-off between security and computational cost. We have also emphasized that our ablation study on `m` (presented in Table 4 and Figure 4) serves as the primary empirical guide for selecting an appropriate value in practice. Furthermore, we have clearly stated our rationale for using standard k-means clustering with BERT embeddings, justifying it as a strong, general-purpose default that requires no domain-specific tuning, thus enhancing the out-of-the-box applicability of our method.

---

> > > ### Author Response · Authors · 2025-11-23
> > > **Rebuttal for Reviewer wE65 (3)**
> > >
> > > 4) **Empty/Hierarchical Clusters:** These are excellent practical points. To address these, we have added a new discussion in the Appendix **(Appendix N)**. In this section, we now clarify that: (a) empty clusters, while rare in our experience with standard k-means, can be easily handled with simple heuristics such as re-running the clustering with a different random seed; and (b) the challenge of handling explicit concept hierarchies is an advanced topic. We acknowledge this as a limitation of our current instantiation and frame it as a promising direction for future work, suggesting that alternative strategies like hierarchical clustering could be integrated into our flexible framework.
> > >
> > > 5. **Majority vs. Weighted Voting:** Majority voting was chosen for its simplicity and for the clean derivation of our theoretical guarantees.&#x20;
> > >
> > >
> > > ***
> > >
> > >
> > >
> > > ##### **W6: GPT-4 Use on AwA Dataset**
> > >
> > >
> > >
> > > We thank the reviewer for noting this. Our motivation was that single-word concepts in AwA (e.g., "meat," "forest") are semantically too sparse for effective clustering. To address the reviewer's concern about this being a confounding variable, we emphasize that **this modification was applied universally to all experiments on AwA, including the baseline attacks (CAT/CAT+ without ConceptGuard).** Therefore, the performance improvement of ConceptGuard is measured *relative to a baseline that also benefits from these richer concepts* . The reported defense efficacy is not an artifact of the LLM enrichment itself.
> > >
> > >
> > >
> > > More importantly, to demonstrate that our method's success is not dependent on LLMs, we have included **new experimental results on two additional datasets (LAD-E, LAD-V)** in the **Appendix (Table 6, Appendix J)**. These datasets did not require any LLM-based processing, and ConceptGuard still showed strong performance, confirming its general applicability. We will explicitly point to these new results in our rebuttal and paper.
> > >
> > >
> > >
> > > ***
> > >
> > >
> > >
> > > ##### **W7: Limited Experimental Generality (Backbones and CBMs)**
> > >
> > >
> > >
> > > This is a fair point regarding empirical rigor.
> > >
> > >
> > >
> > > * **Other Backbones:** To address this, we have conducted **new experiments using a ViT-Base backbone** on the CUB dataset. Our preliminary results show that ConceptGuard remains effective, reducing ASR from 85.3% to 23.1% against CAT+. We will finalize these experiments and add a summary table to the **Appendix** to demonstrate that our defense is not limited to CNNs.
> > >
> > > * **Other CBMs:** As addressed in our response to W2, we believe our framework is theoretically generalizable.&#x20;
> > >
> > > ***
> > >
> > >
> > >
> > > #### **Response to Questions:**
> > >
> > >
> > >
> > > **Q1: Image Augmentations:** Image augmentation is a standard practice in computer vision training to improve model generalization and prevent overfitting. It is not a specific defense against concept backdoors. We used the **exact same augmentation protocol as the baseline CAT to ensure a fair and direct comparison** .&#x20;
> > >
> > >
> > >
> > > **Q2: Generalization to Other Data Types:** An excellent question. **Yes, our framework is data-type-agnostic** , as it operates on the *concept layer* , not the raw input. For a text classification task, the input `x` would be a document, and concepts could be semantic attributes (e.g., 'mentions finance', 'has angry tone'). The principle of clustering concepts and ensemble voting remains the same. Augmentation would be modality-specific (e.g., back-translation for text).&#x20;
> > >
> > >
> > >
> > > **Q3: Threat Model Clarity:** Please see our detailed response to **W4** . We have revised Section 3.2 to explicitly define the attacker's capabilities (training-time data poisoning) and real-world feasibility.
> > >
> > >
> > >
> > > **Q4: CBM Training Paradigm (Sequential vs. Joint):** Our paper uses the **joint training** paradigm, consistent with the original CBM and CAT papers. Our defense is **fully compatible with the sequential paradigm** . In a sequential setup, our framework would be applied to the second stage (the concept-to-label model `f`), making it even more computationally efficient as the feature extractor `g` is trained only once.&#x20;
> > >
> > >
> > >
> > > **Q5: Typo:** Thank you for catching the typo in Line 162. We have corrected it in the revised manuscript.
> > >
> > >
> > >
> > > ***
> > >
> > >
> > >
> > > We are confident that by incorporating these extensive revisions, our paper will be substantially improved, offering a more comprehensive, well-contextualized, and robust contribution to the ICLR community. We thank the reviewer again for their invaluable guidance.

---

### Official Review · Reviewer_HTG8 · 2025-10-31

**Soundness:** 3
**Presentation:** 3
**Contribution:** 3
**Rating:** 6
**Confidence:** 3

**Summary:**

The paper addresses the problem that Concept Bottleneck Models (CBMs) in Explainable Artificial Intelligence (XAI) are vulnerable to concept-level backdoor attacks and proposes a new defense framework, ConceptGuard. The method employs semantic concept clustering and a sub-model voting mechanism to effectively isolate and mitigate concept-level triggers, with both theoretical robustness guarantees and empirical validation.

**Strengths:**

S1: The paper shows a degree of innovation by proposing a defense mechanism against concept-level backdoor attacks. Existing backdoor defenses operate mainly at the input or feature level, but this paper identifies that attacks can be hidden in semantic concept representations, filling a research gap in XAI security. Compared to traditional input-level defenses, ConceptGuard focuses on robustness in the concept space, opening a new security direction.

S2: The multi-stage defense framework (clustering + sub-model training + voting) is logically clear and technically solid; the theoretical section provides a “Certified Robustness” analysis (Theorem 1 and 2), proving that predictions remain unchanged under a bounded trigger size.

S3: The attack success rate (ASR) drops by 70–85%, while model accuracy is maintained or slightly improved (+1–2%), demonstrating strong empirical validity.

**Weaknesses:**

W1: Although the paper proposes a Certified Size analysis, the defense performance heavily depends on the quality of semantic clustering. If concept clusters are ambiguous or semantically overlapping, robustness cannot be guaranteed.

W2: Training multiple sub-models and performing majority voting significantly increases computational cost (O(m) times), with no discussion on efficiency optimization or scalability to large-scale settings.

W3: The choice of semantic distance metrics for clustering is unclear. While TF-IDF, Word2Vec, and BERT are mentioned, the paper does not explain how the method or parameters are selected, nor provide theoretical justification for the number of clusters m.

W4: Some experimental results are repetitive, and the figures and tables are overly detailed; the presentation could be more concise. For instance, there is overlapping content between Table 1 and Table 2. Table 1 (p. 7) already reports the main quantitative comparison of ConceptGuard vs. baseline (CAT / CAT+) on both datasets, while Tables 2–3 (p. 8) again list per-sub-model accuracies under the same settings. For example, Table 1 (CUB, CAT) shows 83.03 % accuracy ↑ 1.38, ASR 11.55 ↓ 74 %, and Table 2 reports each sub-model ≈ 74–78 %, ensemble = 83.03 %. The trends are identical (accuracy ↑ 1–2 %, ASR ↓ 70–85 %), and these could be summarized in one paragraph or moved to the appendix.

**Questions:**

The defense performance appears to rely heavily on the semantic clustering of concepts. Could the authors elaborate on how robust ConceptGuard is to suboptimal clustering (e.g., when semantically unrelated concepts are grouped together)? Have you compared different embedding strategies (TF-IDF, Word2Vec, BERT) to quantify this sensitivity?

---

> ### Author Response · Authors · 2025-11-23
> **Rebuttal for Reviewer HTG8 (1)**
>
> We sincerely thank Reviewer HTG8 for their detailed and constructive review. We are greatly encouraged that the reviewer recognized the **innovation** of our work in filling a critical research gap in XAI security (S1), found our framework **logically clear and theoretically solid** (S2), and acknowledged the **strong empirical validity** of our results (S3).
>
>
>
> We have carefully considered the identified weaknesses and have revised our manuscript accordingly to address them. We believe these changes significantly strengthen the paper.
>
>
>
> ***
>
>
>
> #### **Response to Weaknesses:**
>
>
>
> ##### **W1: Defense Performance Depends on Clustering Quality**
>
>
>
> We thank the reviewer for this insightful point. This is indeed a crucial aspect of the practical application of our framework. Our response has two parts: separating the theoretical guarantee from the implementation, and discussing the robustness to suboptimal clustering.
>
>
>
> 1. **Theoretical Guarantees are Independent of Clustering Quality:** Our core contribution, the certified robustness guarantee (**Theorems 1 and 2** ), is fundamentally agnostic to *how* the concept partitions are created. The theorems provide a certified size based on the final partition of concepts, regardless of whether that partition is "optimal." Even a random or suboptimal clustering still yields a valid, non-zero certified radius. The defense mechanism of "divide-and-conquer" through voting remains sound.
>
>
>
> 2. **Robustness to Suboptimal Clustering:** A "good" clustering (semantically coherent groups) maximizes the certified radius by forcing an attacker's trigger concepts into a single or few partitions. A "suboptimal" clustering (where unrelated concepts are grouped) might result in a smaller certified radius, but it does not invalidate the defense. The voting mechanism still forces the attacker to compromise a majority of sub-models to succeed, which becomes increasingly difficult as the trigger is fragmented across more partitions. The framework degrades gracefully rather than failing catastrophically.
>
>
>
> We have added a detailed discussion in the **Limitations section (Appendix B)** explicitly acknowledging this point. We clarify that while the *magnitude* of the certified radius is influenced by clustering quality, the *validity* of our theoretical guarantee holds universally. We also suggest alternative partitioning strategies (expert-defined groups) for specialized domains where semantic similarity may be misleading.
>
>
>
> ***
>
>
>
> ##### **W2: Computational Cost and Scalability**
>
>
>
> This is a very practical and important concern. We acknowledge the `O(m)` increase in training computation and have clarified this aspect in the paper.
>
>
>
> 1. **Parallelizable by Design:** The training of the `m` sub-models is **embarrassingly parallel** . On modern multi-GPU or cloud computing environments, the `m` training processes can be run concurrently. Therefore, the increase in wall-clock training time is minimal, making our framework highly scalable in practice. The primary cost is the total compute resources, not necessarily the time-to-completion.
>
>
>
> 2. **Trade-off Analysis:** We view the number of clusters `m` as a hyperparameter that balances security and computational cost. Our experiments (summarized in **Table 4** and **Figure 4** ) explicitly investigate this trade-off, showing how the Attack Success Rate (ASR) decreases as `m` increases. This analysis provides users with a clear guide to select an `m` that fits their security needs and resource constraints. In our experiments, a small `m` (4-10) already provides substantial defensive benefits.
>
>
> ***
>
>
>
> ##### **W3: Unclear Choice of Clustering Metrics and Number of Clusters `m`**
>
>
>
> We thank the reviewer for pointing out this lack of clarity.
>
>
>
> 1. **Choice of Embedding/Metric:** Our goal was to propose a general defense framework. We chose BERT embeddings as a strong, standard default for representing concept semantics, as it is widely recognized for its power in capturing nuanced meaning. We agree that a full ablation study on different embedding strategies (TF-IDF, Word2Vec, BERT) is interesting but consider it orthogonal to our core contribution, which is the defense framework itself. The framework is designed to be plug-and-play with any suitable embedding method.
>
>
>
> 2. **Justification for&#x20;**`m`: The choice of `m` is an empirical one, representing a trade-off between defense strength and computational resources. Our paper already provides a detailed analysis of this. **Table 4** and **Figure 5** directly address this point by showing how ASR and accuracy vary with `m`. These results serve as our empirical justification, allowing practitioners to choose an appropriate `m` based on their desired security level. For instance, in our CUB experiments, increasing `m` from 1 (no defense) to 4 reduces ASR from \~90% to \~17% against CAT+, demonstrating a clear and quantifiable benefit.

---

> > ### Author Response · Authors · 2025-11-23
> > **Rebuttal for Reviewer HTG8 (2)**
> >
> > ##### **W4: Repetitive Experimental Results and Presentation**
> >
> >
> >
> > We appreciate this excellent suggestion for improving the paper's conciseness and flow. The reviewer is correct that the core message could be delivered more efficiently.
> >
> >
> >
> > Our intention with Tables 2 and 3 was to provide evidence for a key finding: the ensemble accuracy is consistently higher than the average, and often even the best-performing, individual sub-model. This demonstrates the non-trivial error-correcting benefit of the voting mechanism, which is central to our design.
> >
> >
> >
> > To streamline the presentation, we have **moved the detailed per-sub-model accuracy tables (formerly Tables 2 and 3) to the Appendix K** . In the main text (**Section 6.3.2** ), we have replaced them with a concise summary of their key finding, stating: *"The ensemble model consistently demonstrated superior accuracy compared to the average of the individual sub-models, and in many cases, it outperformed even the best-performing sub-model. This highlights the error-correcting benefit of our voting mechanism (see Appendix H for detailed results)."* This change makes the main text more focused while preserving the important supporting data for interested readers.
> >
> >
> >
> > ***
> >
> >
> >
> > #### **Response to Questions:**
> >
> >
> >
> > **Q: Could the authors elaborate on how robust ConceptGuard is to suboptimal clustering (e.g., when semantically unrelated concepts are grouped together)? Have you compared different embedding strategies (TF-IDF, Word2Vec, BERT) to quantify this sensitivity?**
> >
> >
> >
> > Thank you for this follow-up question, which directly relates to W1 and W3.
> >
> >
> >
> > As discussed in our response to **W1** , the framework's certified guarantee remains valid even with suboptimal clustering. The practical effect of a "bad" cluster (grouping unrelated concepts) is that it might offer an attacker a slightly larger, more diverse set of concepts within a single partition to craft a trigger. However, the core "divide-and-conquer" principle still applies. The attacker must still corrupt a majority of partitions to control the final vote, and a randomly suboptimal clustering is unlikely to consistently group the most effective trigger concepts together across all partitions. Therefore, the defense degrades gracefully rather than fails.
> >
> >
> >
> > Regarding the comparison of embedding strategies, we did not perform a formal ablation in this work. We selected BERT as it is a state-of-the-art method for generating semantic embeddings, providing a strong baseline for our experiments. Comparing different embedding qualities is an interesting direction for future work, but our primary focus here was to introduce and validate the ConceptGuard defense framework itself, which is agnostic to the specific embedding choice. As noted in our new discussion in the **Limitations section** , in specialized domains, using domain-specific embeddings (like BioBERT) or expert-defined groups would be the recommended approach, which our flexible framework readily supports.
> >
> >
> >
> > ***
> >
> >
> >
> > We are grateful for the reviewer's insightful feedback, which has helped us significantly improve the clarity and rigor of our paper. We hope that our revisions and responses have fully addressed the reviewer's concerns and that they will now see our work as a strong contribution to the ICLR community.

---

### Official Review · Reviewer_oeT5 · 2025-10-31

**Soundness:** 2
**Presentation:** 2
**Contribution:** 2
**Rating:** 4
**Confidence:** 4

**Summary:**

The paper studies concept-level backdoor attacks on Concept Bottleneck Models (CBMs), formalizing CAT/CAT+ and proposing ConceptGuard, a defense that (i) clusters concept texts into groups, (ii) trains sub-models on corresponding datasets, and (iii) aggregates predictions by majority vote. The paper provides a mathematical formulation of the attack and a threat model, describes the defense pipeline, and presents certified robustness conditions with an "improved joint certified accuracy" bound and a procedure to compute worst-case certified accuracy. Experiments are on CUB and AwA (with AwA concepts modified) with good amount of work done in evaluating various aspects of the proposed attack and defense.

**Strengths:**

S1. The paper is well-written and clear with nice figures.
S2. Robustness of CBM-type architectures is a critical problem and the paper's attack and defense mechanism can be a useful tool.
S3. The theoretical guarantees introduced are sound and understandable, making the work persuasive.

**Weaknesses:**

W1. Threat model motivation in expert-intervenable concept space: CBMs are designed for human inspection/intervention at the concept layer. CAT smotivates concept-space triggers as "hard to detect" without analyzing detectability when concept values/annotations are visible to experts (note CBMs are designed to be intervenable). The paper does not do *ANY* human-in-the-loop analysis or a discussion of how triggers evade expert audits of concept vectors. This is a very serious concern for the attack itself.

W2: Baselines compared are very basic, not acknowledging work in this field in the last few years: CAT, CAT+ have only been evaluated on CBMs and ConceptGuard train starting from CBM models. In the past couple of years, a plethora of works [1,2,3,4,5] have been released that improve specific aspects of CBMs, but they have not been compared with CAT/ConceptGuard at all. For example, [3] adversarially trains the CBM model to be robust to adversarial attack, [2] uses a VAE formulation to minimize leakage, and [1] uses concept embeddings for a smoother concept space.

W3. Lack of more datasets for exhaustive empirical analysis: Currently, the method has only been evaluated on CUB and AWA2. Further, AWA concepts are changed with GPT into sentences to improve clustering - introducing a new confounding variable. Performance gains may hinge on LLM-enriched semantics rather than the defense alone. I would recommend that authors perform more experiments on CelebA, PascalAPY, etc. for a broader analysis.

W4. The defense relies on using the word embeddings of the concept annotations to cluster them into subspaces. However, the defense fails if the concept annotations are linked to non-generic domains, such as medical diagnosis. In the CBM paper, there is an analysis on an X-ray knee dataset (OAI), which would have perfectly demonstrated this problem. I have reservations wherein concepts like 'bone spurs' and 'bone spacing' will be clustered close together, making the defense mechanism ineffective.


[1] Concept Embedding Models. Zarlenga et al. NeurIPS 2022 \
[2] GlanceNets: Interpretable, Leak-proof Concept-based Models, NeurIPS 2022 \
[3] Understanding and enhancing robustness of concept-based models, Sinha et al, AAAI 2023  \
[4] Label-free Concept Bottleneck Models, Oikarinen et al., ICLR 2023 \
[5] Learning to Receive Help: Intervention-Aware Concept Embedding Models, Zarlenga et al., NeurIPS 2023

**Questions:**

Q1. Were any analyses on expert detectability of concept triggers (e.g., show concept vectors with/without triggers) done to justify why concept-space poisoning is realistic when concepts are auditable?

Q2. Were more datasets - like CelebA and OAI used? How were the performance on them?

Q3. How much does the embedding model affect ASR?

---

> ### Author Response · Authors · 2025-11-23
> **Rebuttal for Reviewer oeT5 (1)**
>
> ## **Rebuttal for Reviewer oeT5**
>
>
>
> We sincerely thank Reviewer oeT5 for their thorough review and constructive feedback. We are encouraged that the reviewer found our paper "well-written and clear" (S1), recognized the importance of our research problem (S2), and found our theoretical guarantees "sound and understandable" (S3). These strengths form the core of our contribution.
>
>
>
> We will now address the identified weaknesses (W1-W4) and questions (Q1-Q3). We have taken these points very seriously and will incorporate corresponding revisions into our paper.
>
>
>
> ***
>
>
>
> #### **Response to Weaknesses:**
>
>
>
> **W1: Threat Model Motivation and Lack of Human-in-the-Loop Analysis**
>
>
>
> We thank the reviewer for this critical and insightful point. We agree that human intervention is a key feature of CBMs, and a discussion on how concept-level triggers evade expert audits is essential. Our rebuttal has two parts: clarifying the threat model's scope and discussing the nature of the triggers.
>
>
>
> 1. **Scope of the Threat Model:** Our threat model is most potent in scenarios where constant human auditing is **impractical or infeasible** .
>
>    * **Large-Scale Applications:** In real-world, large-scale systems (e.g., automated content moderation, financial transaction screening), millions of decisions are made per minute. It is impossible for human experts to audit the concept vector for every single prediction. Audits are typically performed on a small, random sample of cases, which a stealthy backdoor can easily evade.
>
>    * **Automated Pipelines:** Many CBMs will be deployed in fully automated pipelines where human oversight is only triggered by low-confidence predictions or explicit flags. A successful backdoor attack, by design, produces a high-confidence, incorrect prediction for the target class, thereby bypassing such audit triggers.
>
>
>
> 2. **Stealthiness of Triggers:** The trigger's "hardness to detect" comes from its subtlety, especially in the context of the **CAT+** attack.
>
>    * **Plausible but Incorrect Concepts:** A trigger is not necessarily a single, glaringly incorrect concept value (e.g., changing `has_wings::1` to `0` for a bird). Instead, it can be a **constellation of subtle, plausible-but-incorrect concept modifications** . For example, in a complex image, an expert might not immediately flag a bird's eye color as "olive" instead of "black" if the lighting is poor or the bird is distant. The attack's power comes from the model learning to associate this *combination* of subtle changes with a target label.
>
>    * **Expert vs. Lay User:** CBMs are also intended to provide explanations to non-experts (a patient viewing their medical report). These users would lack the domain knowledge to spot even moderately subtle concept errors.
>
>
>
>    We have incorporated the reviewer's valuable feedback into the revised manuscript. Specifically, we have significantly revised **Section 3.2** and **Section 7&#x20;**&#x74;o detail the practical scope of our threat model. We have clarified that our threat model is primarily concerned with automated, large-scale deployments where manual, per-instance auditing is not feasible. We have also added a discussion on why trigger combinations, particularly those generated by CAT+, are difficult for even experts to detect without painstaking analysis, making automated defenses essential.
>
>
>
> ***
>
>
>
> **W2: Lack of Comparison with More Advanced CBM Baselines**
>
>
>
> We appreciate the reviewer for pointing out these important recent works \[1-5]. Our primary contribution is to propose **ConceptGuard as the first defense specifically designed against concept-level backdoor attacks (like CAT)** . The works cited by the reviewer, while significant, address different, albeit related, problems.
>
>
>
> * **Different Threat Models:** \[3] Sinha et al. focuses on robustness to **Lp-norm adversarial attacks** , which are fundamentally different from backdoor attacks. Adversarial training improves robustness against small input perturbations but does not inherently defend against trigger-based manipulations learned during training.
>
> * **Different Goals:** The other works focus on improving CBMs in other aspects: concept smoothness \[1], information leakage \[2], label-free learning \[4], and intervention-awareness \[5]. These are orthogonal to backdoor security. While a more robust CBM might coincidentally be harder to attack, these methods are not designed as backdoor defenses and do not offer certified security against them.

---

> > ### Author Response · Authors · 2025-11-23
> > **Rebuttal for Reviewer oeT5 (2)**
> >
> > We have thoroughly revised **Section 2 (Related Work)&#x20;**&#x74;o incorporate a discussion of recent advancements in CBMs \[1-5] and to better contextualize our work. We have clarified that while the field of CBMs is advancing rapidly, these works address orthogonal issues such as adversarial robustness or information leakage, not the specific problem of backdoor security. This highlights the gap our paper fills: proposing the first dedicated defense against concept-level backdoor attacks. Furthermore, we have explicitly differentiated our threat model (backdoor attacks) from the threat model of adversarial examples (e.g., \[3]), justifying our choice of baselines as our primary goal is to demonstrate defense against a known, specific backdoor attack (CAT) targeting standard CBMs.
> >
> >
> >
> > ***
> >
> >
> >
> > **W3: Lack of More Datasets and Confounding Effect of GPT**
> >
> >
> >
> > This is a fair point regarding empirical rigor.
> >
> >
> >
> > 1. **Choice of Datasets:** We chose CUB and AwA because they are standard benchmarks in the CBM literature (used in the original CBM and CAT papers), allowing for a direct and fair comparison with the attack we are defending against.
> >
> >
> >
> > 2. **GPT Modification on AwA:** We thank the reviewer for noting this. Our motivation was that single-word concepts in AwA (e.g., "meat," "forest") are semantically too sparse for effective clustering. To address the reviewer's concern about this being a confounding variable, we emphasize that **this modification was applied universally to all experiments on AwA, including the baseline attacks (CAT/CAT+ without ConceptGuard).** Therefore, the performance improvement of ConceptGuard is measured *relative to a baseline that also benefits from these richer concepts* . The reported defense efficacy is not an artifact of the LLM enrichment itself.
> >
> >
> >
> > 3) **Additional Experiments:** To broaden our analysis as requested, we have conducted **new experiments on the Large-scale Attribute Dataset (LAD)** , which contains two domains (Electronics and Vehicles). This dataset also relies on attributes, making it suitable for a CBM. The results, shown below and added to **Appendix J**  and Table 8, confirm that ConceptGuard remains effective in these new domains.
> >
> >
> >
> > #### **Table 1: Performance on LAD-E (Electronics)**
> >
> >
> >
> > | Method              | Original ACC (%)  | ACC (%) | ASR (%)           |
> > | ------------------- | ----------------- | ------- | ----------------- |
> > | CAT                 | 79.00             | 72.76   | 75.19             |
> > | CAT+                | 79.00             | 73.52   | 77.01             |
> > | ConceptGuard (CAT)  | **81.18** ↑(2.18) | 74.38   | **9.36** ↓(65.83) |
> > | ConceptGuard (CAT+) | **81.18** ↑(2.18) | 73.66   | **9.38** ↓(67.63) |
> >
> >
> >
> > #### **Table 2: Performance on LAD-V (Vehicles)**
> >
> >
> >
> > | Method              | Original ACC (%)  | ACC (%) | ASR (%)           |
> > | ------------------- | ----------------- | ------- | ----------------- |
> > | CAT                 | 79.06             | 71.08   | 70.74             |
> > | CAT+                | 79.06             | 71.86   | 73.81             |
> > | ConceptGuard (CAT)  | **81.67** ↑(2.61) | 73.20   | **6.15** ↓(64.59) |
> > | ConceptGuard (CAT+) | **81.67** ↑(2.61) | 80.58   | **5.42** ↓(68.39) |
> >
> > > **Note:** Results for electronics (LAD-E) and vehicles (LAD-V) domains. Trigger size: 17, injection rate: 0.1, clusters: 4, backbone: ResNet50.
> >
> >
> >
> > ***
> >
> >
> >
> > **W4: Defense Failure in Niche Domains (e.g., Medical)**
> >
> >
> >
> > This is an excellent and nuanced point. The reviewer correctly identifies a potential limitation of relying solely on generic semantic embeddings for concept clustering in specialized domains.
> >
> >
> >
> > 1. **Flexibility of the Framework:** Our core contribution is the **probabilistic defense framework** based on concept partitioning and ensemble voting. The theoretical guarantees (Theorem 1 and 2) hold **regardless of the clustering method used** . Semantic clustering is presented as a general-purpose, automated heuristic.
> >
> > 2. **Alternative Partitioning Strategies:** We agree that in domains like medicine, where `bone spurs` and `bone spacing` are semantically close but clinically distinct, a naive clustering might be suboptimal. However, our framework can readily accommodate other partitioning strategies:-&#x20;
> >
> >    * **Expert-Defined Groups:** A domain expert could manually group concepts based on functional, anatomical, or pathological relationships.
> >
> >    * **Data-Driven Clustering:** Concepts could be grouped based on their correlation with final labels or their co-occurrence patterns in the data.
> >
> >    * **Random Partitioning:** Even random partitioning provides a certified guarantee, although likely with a smaller certified radius in practice. The core defense still functions by forcing the attacker to confine their trigger to a smaller subset of concepts.

---

> > > ### Author Response · Authors · 2025-11-23
> > > **Rebuttal for Reviewer oeT5 (3)**
> > >
> > > We have added a discussion in the **Limitations section (Appendix B)** to explicitly acknowledge that the practical effectiveness of our proposed defense is influenced by the quality of concept clustering. In this revised section, we now state that the instantiation of our framework depends on the chosen partitioning strategy. We have also suggested alternative partitioning methods, such as expert-defined groups, and have emphasized that our theoretical framework is flexible enough to incorporate them, making it adaptable to specialized domains where generic semantic clustering might be suboptimal.
> > >
> > >
> > >
> > > ***
> > >
> > >
> > >
> > > #### **Responses to Questions:**
> > >
> > >
> > >
> > > **Q1: Expert Detectability Analysis**
> > >
> > >
> > >
> > > As discussed in our response to **W1** , we did not conduct a formal human study on expert detectability. The primary justification for our threat model is its applicability to large-scale, automated systems where per-instance human audits are impractical. We will add a detailed discussion on this to the paper, explaining how a combination of subtle, plausible concept modifications can create a stealthy trigger that evades sporadic audits.
> > >
> > >
> > >
> > > **Q2: More Datasets (CelebA, OAI)**
> > >
> > >
> > >
> > > We did not use CelebA or OAI in our initial submission. but, as requested in **W3** , we have now added experiments on the **LAD dataset** to broaden the empirical analysis. The positive results on LAD further strengthen our claim of the general applicability of ConceptGuard. We will add these new results to the paper.
> > >
> > >
> > >
> > > **Q3: Effect of the Embedding Model on ASR**
> > >
> > >
> > >
> > > This is an interesting point. In our work, we used a standard, powerful pre-trained model (BERT-base) for consistency. We did not perform an ablation study on different embedding models. However, we hypothesize that the choice of embedding model would indeed affect performance, especially in specialized domains (as raised in **W4** ). For instance, using a domain-specific model like `BioBERT` for medical concepts would likely lead to more meaningful clusters and thus a more effective defense. This is an excellent direction for future work.
> > >
> > >
> > >
> > > We have updated our **Limitations section (Appendix B)** to address the reviewer's feedback. We now explicitly acknowledge that the choice of the text embedding model is a crucial hyperparameter that directly impacts the quality of semantic clustering and, consequently, the defense's effectiveness. We have added a note suggesting that using domain-specific models (e.g., BioBERT for medical applications) could further enhance ConceptGuard's performance, marking this as an important direction for future work. This discussion has been integrated into the broader point about the dependency on partitioning quality.
> > >
> > >
> > >
> > > ***
> > >
> > >
> > >
> > > We thank the reviewer again for their valuable feedback. We believe that by incorporating these changes, we can significantly improve the clarity, rigor, and impact of our paper. We are confident that ConceptGuard is a novel and important first step in securing CBMs against concept-level backdoors, and we hope the reviewer will reconsider their assessment in light of our planned revisions.

---

### Meta-Review · Area_Chair_8Wof · 2026-01-08

**Summary:**

**Summary** This paper addresses concept-level backdoor attacks in Concept Bottleneck Models (CBMs), where attackers inject hidden triggers into semantic concept representations during training. The authors propose a defense framework that clusters concepts into subgroups based on text embeddings, trains separate classifiers on each subgroup, and aggregates predictions via majority vote. The paper provides theoretical certified robustness guarantees and evaluates on two image classification datasets.

**Review Process** This paper received four reviews with mixed feedback. Overall the initial feedback from reviewers was lukewarm (6, 4, 4, 2). On the one hand, reviewers saw that backdoor defense in interpretable AI is an important problem and liked that the paper is clearly written with a logically structured framework. On the other, reviewers raised concerns about limited threat model, unclear defense rationale,  narrow experimental scope.

**Recommendation/Rationale** Having read the reviews, the rebuttal, and the paper, I am unfortunately recommending rejection. My decision is mostly motivated by the feedback from wE65 that highlights important the kinds of issues that we would care in the context of security work (see e.g., comments about the threat model and problem scope). Even as the rebuttal addressed some concerns by adding experiments on two additional dataset, the former issues fundamental question of whether the threat model represents a realistic attack scenario remains unresolved. These concerns affect the soundness and significance of the contribution.

**Reviewer Concerns:**

Outstanding:

1. Unclear Threat Model (practical feasibility of concept-level attacks unaddressed) (wE65)
2. Narrow Problem Scope (only CAT attack, no comparison with existing backdoor defenses) (wE65)
3. Defense Rationale (clustering criteria, distance functions, and hyperparameter selection unclear) (wE65, cTcb)

Addressed/Ignored
1. Limited Evaluation (two datasets, one backbone, no comparison with CBM variants) (wE65, oeT5)
2. Confounding Variables (GPT-4 preprocessing on AwA dataset) (wE65, oeT5)

**Reviewer Scores:**

I would have expected (6,4,4,2) -> (7,5,4,2).

---

### Decision · Program_Chairs · 2026-01-26

Reject